# Determinants of transcription factor regulatory range

Chen-Hao Chen[1,2,3], Rongbin Zheng[4], Collin Tokheim[1,3], Xin Dong[4], Jingyu Fan[4], Changxin Wan[4], Qin Tang[3,5], Myles Brown[3,5], Jun S. Liu[6], Clifford A. Meyer[1,3✉] & X. Shirley Liu[1,3,6✉]

Characterization of the genomic distances over which transcription factor (TF) binding influences gene expression is important for inferring target genes from TF chromatin immunoprecipitation followed by sequencing (ChIP-seq) data. Here we systematically examine the relationship between thousands of TF and histone modification ChIP-seq data sets with thousands of gene expression profiles. We develop a model for integrating these data, which reveals two classes of TFs with distinct ranges of regulatory influence, chromatin-binding preferences, and auto-regulatory properties. We find that the regulatory range of the same TF bound within different topologically associating domains (TADs) depend on intrinsic TAD properties such as local gene density and G/C content, but also on the TAD chromatin states. Our results suggest that considering TF type, binding distance to gene locus, as well as chromatin context is important in identifying implicated TFs from GWAS SNPs.

[1] Department of Data Sciences, Dana-Farber Cancer Institute. Harvard T.H. Chan School of Public Health, Boston, MA, USA. [2] Biological and Biomedical Science Program, Harvard Medical School, Boston, MA, USA. [3] Center for Functional Cancer Epigenetics, Dana-Farber Cancer Institute, Boston, MA, USA. [4] Clinical Translational Research Center, Shanghai Pulmonary Hospital, School of Life Sciences and Technology, Tongji University, Shanghai 200092, China. [5] Department of Medical Oncology, Dana-Farber Cancer Institute, Harvard Medical School, Boston, MA, USA. [6] Department of Statistics, Harvard University, Cambridge, MA, USA. ✉email: cliff_meyer@ds.dfci.harvard.edu; xsliu@ds.dfci.harvard.edu

hIP-seq is broadly used for identifying the genome-wide binding sites of specific TFs[1,2]. Thousands of these binding-site profiles, or cistromes[3], have been produced in cells and tissues[3]. One important use of cistromes is to identify the TFs that regulate a given gene. However, it is difficult to assign most TF-binding events to a gene (or vice versa) because relatively few TF-binding sites occur very near or within a gene promoter[3]. In most studies, target genes are designated using ad hoc methods, such as assigning TF-binding sites to the nearest gene or to genes within arbitrary genomic distance thresholds. More accurate approaches[4] use soft thresholds and consider the effect of multiple binding sites, but still use arbitrarily defined parameters and model all genes with the same parameters[4]. More recently, genome-wide chromatin conformation Hi-C maps have revealed that the genome is organized as a hierarchically nested structure. Topologically associating domains (TADs)[5] represent one level of this hierarchy that influence enhancer activity by facilitating enhancer-promoter interactions within TADs[5]. However, Hi-C maps detect fewer enhancer-promoter interactions than expected based on known regulatory interactions[6]. How to use Hi-C observations in quantitative models of cis-regulatory interactions remains an open question. In principle, systematic analysis of Hi-C, TF cistromes, histone mark ChIP-seq, and gene expression data could reveal insights into gene regulatory mechanisms that lead to better predictions of TF target genes and, in turn, more accurate interpretation of non-coding GWAS hits.

In this study, we systematically model the genomic distance over which TFs regulate genes, and evaluate how these regulatory ranges depend on TFs and on genomic and chromatin contexts (measured by H3K27ac ChIP-seq[7]). Our integrative analyses of large compendia of ChIP-seq[3], gene expression[8,9], and eQTL[9] data reveal a previously undescribed relationship between TF regulatory ranges and local genomic and chromatin contexts. These results suggest the existence of two distinct 'short-range' and 'long-range' classes of TF with contrasting characteristics, including genome-wide binding, auto-regulatory, tissue-restricted expression, and pioneer-factor like properties. In addition, the regulatory ranges of long-range TFs are longer in repressed genomic regions than in more active ones, suggesting the wide-spread existence of enhancers that regulate genes up to a hundred kb away in gene-sparse regions. Finally, we apply our findings to identify TFs implicated with disease-associated SNPs. Unlike previous studies, which mostly focus on individual TFs, our findings illustrate commonalities between TFs and provided a succinct framework for interpreting complex gene regulation.

## Results

**Distance-based model of TF binding and gene expression.** TF cistromes produced by ChIP-seq represent the genome-wide locations of TF-binding sites and potential cis-regulatory elements. To infer the gene regulatory characteristics of different TFs, we used the regulatory potential (RP) model[10] to describe the relationships between TF cistromes and gene expression (Fig. 1a). The RP model considers contributions of all binding sites of TF $i$ near the transcription start site (TSS) of gene $j$ by summing the regulatory effect of individual binding sites weighted by binding-site-to-TSS distances. This model is based on two assumptions consistent with experimental observations. First, a TF-binding site's regulatory effects on a gene's expression level typically diminishes monotonically with the genomic distance separating the TF-binding site and the gene TSS. This assumption is motivated by the genomic distance-dependent monotonic decay observed in chromatin interaction experiments[11], eQTL studies showing that SNPs associating with the gene expression

often fall close to the gene[12], and CRISPR enhancer screens showing perturbations affecting expression are enriched near the TSS[13]. The second assumption is that the contribution of each enhancer is independent of the others so that the influence of multiple enhancers on a gene is represented by the sum of their individual contributions. With these assumptions, we used an exponential decay function, parameterized by a "decay distance" constant $\Delta$, within which distance the TF regulatory effects are halved. We calculated the RP score, $R_{i,j}(\Delta)$, using the sum of TF $i$ binding sites weighted by their distance to the TSS of gene $j$ with an exponential decay distance constant ($\Delta$), to estimate the regulatory effect of TF $i$ on gene $j$. (Fig. 1a; See methods for statistical details).

Although the RP model does not encompass the full complexity of gene regulation, it does allow us to quantify the genomic distances over which TFs influence genes. When the decay distance ($\Delta$) is small, only TF-binding sites near the TSS contribute to the regulation of the gene; when the decay distance ($\Delta$) is large, distant binding sites contribute. In a gene expression experiment involving the perturbation of the expression of a single TF, comparison of mRNAs before and shortly after the perturbation reveals the differentially expressed (DE) genes that are likely to be directly regulated by that TF. If ChIP-seq data for a TF was available under the same experimental conditions, the RP model can be applied to predict the likely TF target genes from the ChIP-seq peaks. It is important to note that $R_{i,j}(\Delta)$ depends on the decay distance, $\Delta$, and changing the decay distance will affect the ranking of modeled TF $i$ regulatory effects on gene $j$. Systematically varying $\Delta$ across a broad range we can determine the $\Delta$ that best separates the DE gene set from the other genes by calculating the Kolmogorov–Smirnov statistic for each of the tested $\Delta$ value. We refer to the optimal decay distance as $\Delta_i^*$, the "regulatory decay distance" of TF $i$. Using this approach, we estimated $\Delta_i^*$ for the sets of genes that are downregulated on knockdown of MYC[14], FOXM1[15], GABPA[16], ESR1[17], or upregulated on dihydrotestosterone stimulation of AR[18] and on dexamethasone stimulation of NR3C1[19] (Fig. 1b, Supplementary Fig. 1a). In all cases the associations between the differentially expressed gene sets and the peak RPs were highly significant. We observed different regulatory decay distances $\left(\Delta_i^*\right)$ for different TFs. The $\Delta_i^*$ for MYC, GABPA, and FOXM1 were shorter ($\Delta_i^* < 1$ kb) than for AR, ESR1, and NR3C1 ($\Delta_i^* > 10$ kb). Besides, a comparison of the RP model with a model that predicts target genes from TF ChIP-seq on the basis of the distance from the TSS to the nearest peak shows that the RP model better predicts target genes (Supplementary Fig. 1b).

**Different regulatory distances define two classes of TF.** TF perturbation data sets with DE genes and TF ChIP-seq data generated in the same cell type are scarce. To infer the regulatory decay distances ($\Delta_i^*$) for more TFs, we used gene expression to identify likely TF target genes. We used correlations between gene expression levels of TFs and other genes across cell lines or tissues, reasoning that the TF expression would better correlate with expression of genes regulated by that TF than those not regulated by the TF. We estimated the $\Delta_i^*$ to be the $\Delta$ value that maximizes the agreement between target genes calculated from TF $i$ binding sites and target genes calculated from TF $i$ gene expression correlations (Fig. 1c; methods). With $\Delta_i^*$, the association between the RP scores of TF $i$ on gene $j$ ($R_{i,j}(\Delta_i^*)$) and the correlations of TF $i$ with gene $j$ ($\rho_{i,j}^{\text{expr}}$) is highly significant ($p$ value $< 10^{-8}$) for all TFs included in this study (Supplementary Fig. 2a).

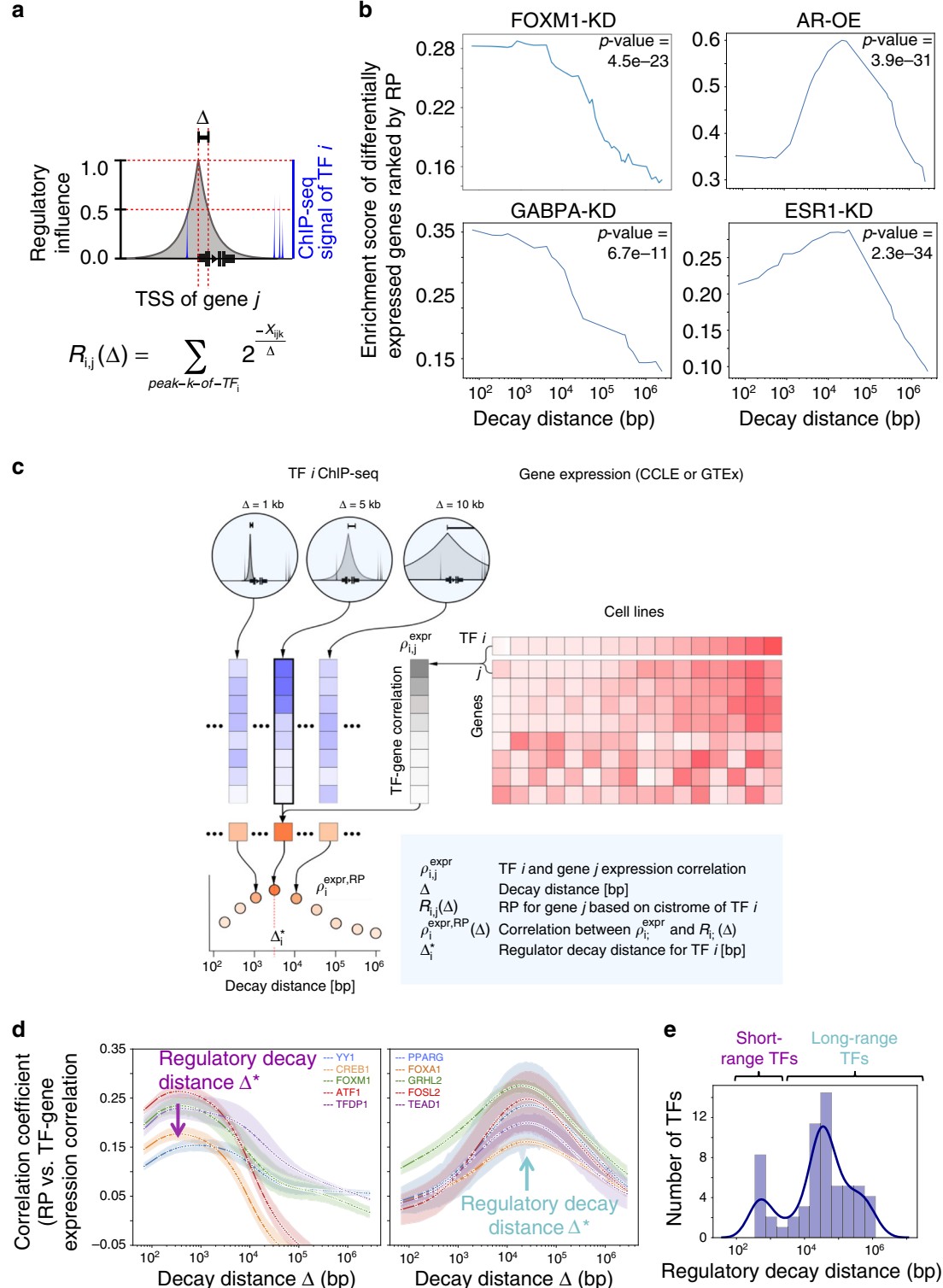

We first inferred the $\Delta_i^*$ using the Cistrome DB TF ChIP-seq collection[3] and gene expression data across ~1000 cell lines from the Cancer Cell Line Encyclopedia (CCLE). $\Delta_i^*$ is <1 kb for some TFs such as *YY1*, *CREB1*, *FOXM1*, *ATF1*, and *TFDP1* (Fig. 1d left, Supplementary Fig. 2b left), but is >10 kb for TFs such as *PPARG*, *FOXA1*, *GRHL2*, *FOSL2*, and *TEAD1* (Fig. 1d right, Supplementary Fig. 2b right). We repeated the analysis above using another five gene expression cohorts[20–24], and further used Genotype-Tissue Expression (GTEx)[9] to infer tissue-specific $\Delta_i^*$ (methods).

These analysis yielded similar estimates of $\Delta_i^*$ across different gene expression cohorts using both linear and nonlinear approaches (Methods, Supplementary Fig. 2d–e). Moreover, for the same TF, the inferred $\Delta_i^*$ is similar in different tissues (Supplementary Fig. 2f). Of 108 TFs with at least three ChIP-seq samples containing at least 20,000 peaks in Cistrome DB[3], we identified 60 TFs with at least three ChIP-seq samples that yielded significant $\Delta_i^*$ estimates in at least three gene expression cohorts (Supplementary Data 1). We observed a distribution of $\Delta_i^*$

**Fig. 1 RP mode reveals two distinct TF classes: short-range and long-range. a** Schematic of the regulatory potential (RP) model. The regulatory effect of TF $i$ on gene $j$ is modeled as the RP, R$i$, $j(\Delta)$, which sums up all TF $i$ ChIP-seq binding effects on the gene $j$. The effect of a single binding site $k$ of TF $i$ on gene $j$ decays exponentially with increasing $x_{ijk}$, the genomic distance between TSS of gene $j$ and TF $i$ binding site $k$. The exponential decay function ($2^{\frac{-x_{ijk}}{\Delta}}$) is parameterized by the decay distance $\Delta$, the distance at which the TF regulatory effects are halved. **b** TF $i$-specific regulatory decay distances ($\Delta_i^*$) can be inferred as the $\Delta$ that best separates TF $i$ perturbation-induced differentially expressed (DE) genes from other genes. $R_{i,j}(\Delta)$ with short-range (<1 kb) best separates FOXM1-knockdown or GABPA-knockdown DE gene sets (left). AR overexpression or ESR1-knockdown DE gene sets are best separated by $R_{i,j}(\Delta)$ with long-range $\Delta$ (>10 kb). The two-sided Kolmogorov–Smirnov two-sample test is used to estimate the degree of separation of DE genes from other genes. **c** $\Delta_i^*$ can also be inferred as the $\Delta$ that leads to the best concordance between TF $i$ regulatory effects estimated by TF $i$ ChIP-seq ($R_{i,j}(\Delta)$) and expression cohorts ($\rho_{i,j}^{\mathrm{expr}}$: TF $i$-gene $j$ expression correlations), respectively. A second correlation coefficient $\rho_i^{\mathrm{expr,RP}(\Delta)}$ was calculated to measure the concordance between $\rho_{i,j}^{\mathrm{expr}}$ and $R_{i,j}(\Delta)$ (see the main text for the rationale and Methods for statistical details). **d** TFs with short-range $\Delta_i^*$ (100bp-3 kb) include YY1, CREB1, FOXM1, ATF1, and TFDP1 (left). TFs with long-range $\Delta_i^*$ (3 kb–100 kb) include PPARG, FOXA1, GRHL2, FOSL2, and TEAD1 (right). Colored shaded regions depict the 95% confidence intervals derived from all ChIP-seq samples that passed QC for each TF. Dots along the line are $\Delta$ values being tried. **e** Distribution of regulatory decay distances ($\Delta_i^*$) of 11 short-range TFs (left) and 49 long-range TFs (right). Source data are provided as a Source Data file.

(Fig. 1e, Supplementary Fig. 2c) with two strong modes, suggesting two distinct classes of TF: short-range TFs (100 bp to 3 kb) and long-range TFs (>3 kb). Although there might exist more than two modes, the following two-class analysis remains useful in contrasting the properties of short- and long-range TFs.

Compared with the cistromes of long-range TFs, those of short-range TFs are significantly enriched within 1 kb of TSSs (Supplementary Fig. 2g) and CpG islands (Supplementary Fig. 2h). This indicates that the distinct regulatory ranges observed in our model recapitulate the biological properties of traditionally defined promoters and enhancers.

**Long- and short-range TFs have distinct properties**. We investigated other properties of short- and long-range TFs. First, we wondered whether different groups of TFs preferentially bind to genomic regions with different levels of H3K27ac, a marker of active promoters and enhancers[7]. To define genomic intervals likely to be of relevance to transcriptional regulation, we selected Hi-C-derived TADs as genomic units. We characterized TADs using mean H3K27ac ChIP-seq read density from 1545 human H3K27ac ChIP-seq samples across diverse cell types in the Cistrome DB[3] and examined their signals across 3051 previously defined TADs[5] (Supplementary Data 2). We clustered TADs across samples using the mean H3K27ac signal, revealing coordinated TAD usage in different cell lineages (Fig. 2a, top). Genes within TAD clusters were enriched in gene ontology categories relevant to the tissue clusters[25] (Supplementary Data 3, 4). By mapping the TAD clusters to A/B compartments annotated in 21 Hi-C samples[26], we found the H3K27ac level corresponds with the percentage of compartment A (Fig. 2a, bottom). Therefore, we defined A-type TADs (316 TADs) and B-type TADs (649 TADs), corresponding to high and low H3K27ac signal across most samples, respectively (Fig. 2a, top). B-type TADs have more A/T-rich DNA[27], lower gene density, longer gene transcripts, and more tissue-restricted gene expression (Supplementary Fig. 3a–b).

Having characterized the genomic regions with differential H3K27ac marks, we went on to examine the binding density of 1396 TF ChIP-seq samples. In general, TFs prefer to bind to A-type TADs (Supplementary Data 5, Fig. 2b, top). To investigate whether regulation within a given TAD could be dominated by a given TF, we calculated a $z$ score for each TF-TAD pair, reflecting the binding density of this TF in the TAD relative to the binding densities of other TFs in the same TAD. Certain TFs, such as *YY1*, favor A-type TADs, while others, such as *TEAD1*, favor B-type TADs (Fig. 2b, bottom). In general, the short-range TFs tend to have higher $z$ scores in the A-type TADs, whereas a large proportion of long-range TFs tend to prefer the B-type TADs (Supplementary Fig. 3c). This finding is consistent with well-known principles in which lineage-specifying genes are subject to

more long-range regulation, whereas ubiquitously expressed genes are regulated at closer range.

Previous studies suggested that TADs partition the genome into functional domains that help coordinate gene expression[28], and genes with related functions are occasionally clustered together[29] and expressed in the same tissues[30]. We reasoned that if a set of TADs were heavily bound by one long-range TF, its long-range regulatory properties would enable that TF to regulate a significant proportion of the genes located within those TADs. We defined a TF's "target" TADs as those TADs that are densely bound by the given TF relative to other TFs in the Cistrome DB ($z$ score > 1, Fig. 2c). An example of a *TEAD1* target TAD (Fig. 2d top) demonstrates the nearly perfect colocalization of *TEAD1* and H3K27ac ChIP-seq peaks, suggesting that *TEAD1* dominates the activity of its target TAD. Indeed, the GO term enrichment of genes within the target TADs correspond to the known functions of the dominating TFs (Supplementary Data 6).

To investigate whether long-range TFs may influence the chromatin state of target TADs as predicted, we examined whether the expression of long-range TFs are robustly associated with H3K27ac levels in the target TADs. For each TF, we selected CCLE cell lines with available H3K27ac ChIP-seq data and sorted these cell lines by the TF expression level. We observed positive associations between the TF expression and the mean H3K27ac levels of its target TADs for most long-range TFs such as TEAD1 (Fig. 3a–b), but weaker or negative associations for short-range TFs such as YY1 (Fig. 3a–b). Similarly, such significant positive associations hold between the expression of long-range TFs, such as *TEAD1*, and genes located in their target TADs (Fig. 3c–d, Supplementary Fig. 3d) and weaker associations for short-range TFs. These contrasting results between long- and short-range TFs suggest that long-range TFs might influence the chromatin state and gene expression across their target TADs than short-range TFs.

As lineage-specifying genes are subject to more long-range regulation, we assessed whether the long-range TFs are expressed in a lineage-specific manner. As predicted, their expression across 48 GTEx tissues shows that long-range TFs are generally expressed in a tissue-restricted fashion, reflected by their higher coefficients of variation across GTEx tissues (Fig. 3e). Self-activating TF comprised positive feedback loops have been proposed as a mechanism for establishing stable gene expression programs during lineage specification[31]. Indeed, compared with short-range TFs, TADs that harbor long-range TF genes are heavily bound by these TFs themselves ($p$ value = 0.008), suggesting that long-range TFs tend to have auto-regulatory properties (Fig. 3f, Supplementary Fig. 3e). This indicates that multiple binding sites of the same TF within the TAD harboring the TF gene itself may serve as a robust auto-regulatory mechanism for maintaining lineage restricted long-range TF expression.

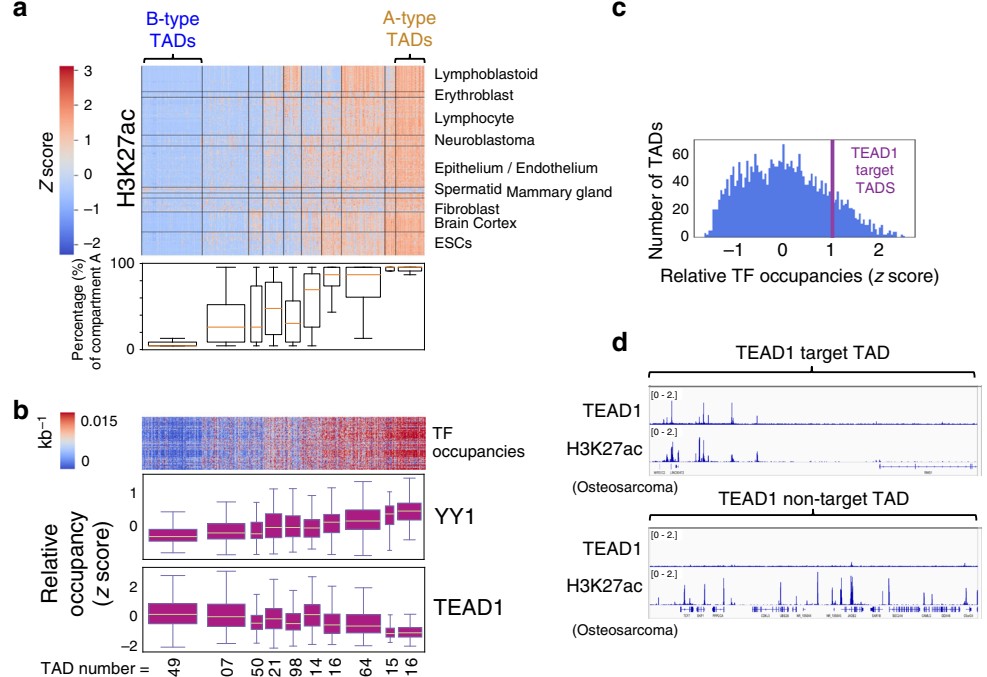

**Fig. 2 Long-range TFs and short-range TFs have distinct binding profiles. a** Top: heatmap of average H3K27ac levels in 3051 TADs across 1544 samples. Hierarchical clustering derives 10 TAD clusters and 10 sample clusters, with TAD clusters re-ordered using mean H3K27ac abundances. 316 A-type TADs and 649 B-type TADs correspond to high and low H3K27ac levels in the majority of samples. Bottom: percentage of Hi-C derived compartment A in TAD clusters. The box plot extends from the lower to the upper quartile values of the data, with a line at the median. The whiskers extend from the box to show the range of the data. **b** Top: ChIP-seq peak numbers per kb in TADs across 1396 samples. For each TF ChIP-seq sample only the most significant 20,000 peaks are used, and the TADs are ordered as in Fig. 2a. Bottom: the relative occupancies ($z$ scores) of YY1 and TEAD1 in TAD clusters. The relative occupancy of TF $i$ in a given TAD is the binding density of the TF in that TAD relative to the binding densities of other TFs in the same TAD. The box plot illustrates data as **a**. **c** TEAD1 target TADs are defined as those TADs with a TEAD1 occupancy ≥1 standard deviation among all TFs ($z$ scores ≥ 1). **d** In SF269 cell, TEAD1 ChIP-seq peaks and H3K27ac signals colocalize well in a TEAD1 target TAD, but not in a non-target one. Source data are provided as a Source Data file.

Having shown that a large proportion of long-range TFs bind to B-type TADs, we reasoned the abundant long-range TF binding within these TADs might require pioneer-like properties, characterized by the capability to facilitate the binding of other TFs at their binding sites[32]. In the CCLE cell lines, we evaluated whether a TF's DNA motif becomes more enriched in the ChIP-seq peaks of other factors as the TF's expression increases. For instance, in cell lines with increased *TEAD1* expression, the *TEAD1* motif becomes more enriched in the ChIP-seq peaks of other factors (Fig. 3g). TFs with such "pioneer-factor-like" properties are generally long-range TFs (Fig. 3h, Methods, Supplementary Data 7), including known pioneer factors (*FOXA1*[32], *GRHL2*[33]) and previously unknown ones (*FOSL2*) (Supplementary Fig. 3f).

**Regulatory distances differ between A- and B-type TADs.** Having shown that the regulatory distances are TF specific, we investigated other determinants. In the RP model, we have assumed that for a given TF the same regulatory distance $\Delta_i^*$ applies to all target genes and found the modal regulatory distance of long-range TFs to be 47 kb (Fig. 1e). However, several genes, including *BCL11A*[34], *GATA3*[35], *PAX6*[36], *SHH*[37], and *SOX9*[38], are regulated by enhancers over much longer genomic distances. Notably, these genes occur in genomic regions that have low gene densities and low gene expression levels. We hypothesized that there exist systematic differences in TF regulatory decay distances between active and repressed genomic regions.

Having defined A-type and B-type TADs with high and low H3K27ac marks, respectively (Fig. 2a), we examined whether regulatory distances differ between these TAD types. Using the RP model while restricting the analysis to genes located within A-type and B-type TADs, respectively, we calculated TAD type-specific regulatory decay distances, $\Delta_i^A$ (in A-type TADs) and $\Delta_i^B$ (in B-type TADs) for each TF $i$. For long-range TFs, their regulatory decay distances are longer in the B-type TADs (i.e. $\Delta_i^B > \Delta_i^A$) (Fig. 4a), suggesting the few documented long-range enhancer activity in B-type TADs being the norm rather than the exception[35]. We obtained consistent results using CCLE gene expression data and tissue-matched GTEx data (Fig. 4b; Supplementary Fig. 4a). Although TF-binding sites and TSSs tend to be sparser in B-type TADs, the model is not merely reflecting peak to gene distance distributions. An analysis of regulatory decay distances inferred using the most significant 10,000 and 20,000 peaks produces almost the same results (Supplementary Fig. 4b), whereas the peak-to-TSS distances are inversely related to peak numbers (Supplementary Fig. 4c).

We went on to verify whether enhancers regulate genes over longer genomic distances in B-type TADs than in A-type ones using complementary data. CAGE-seq[39], which captures the TSSs of both mRNAs and eRNAs, allows us to compute the transcription correlation between coding genes and eRNAs as a function of their separation distances. We found higher enhancer-promoter correlations over longer genomic distances in B-type TADs (Supplementary Fig. 4d). Analysis of GTEx eQTL data revealed the distances between eQTLs and the corresponding gene TSSs to be significantly longer in B-type TADs in most

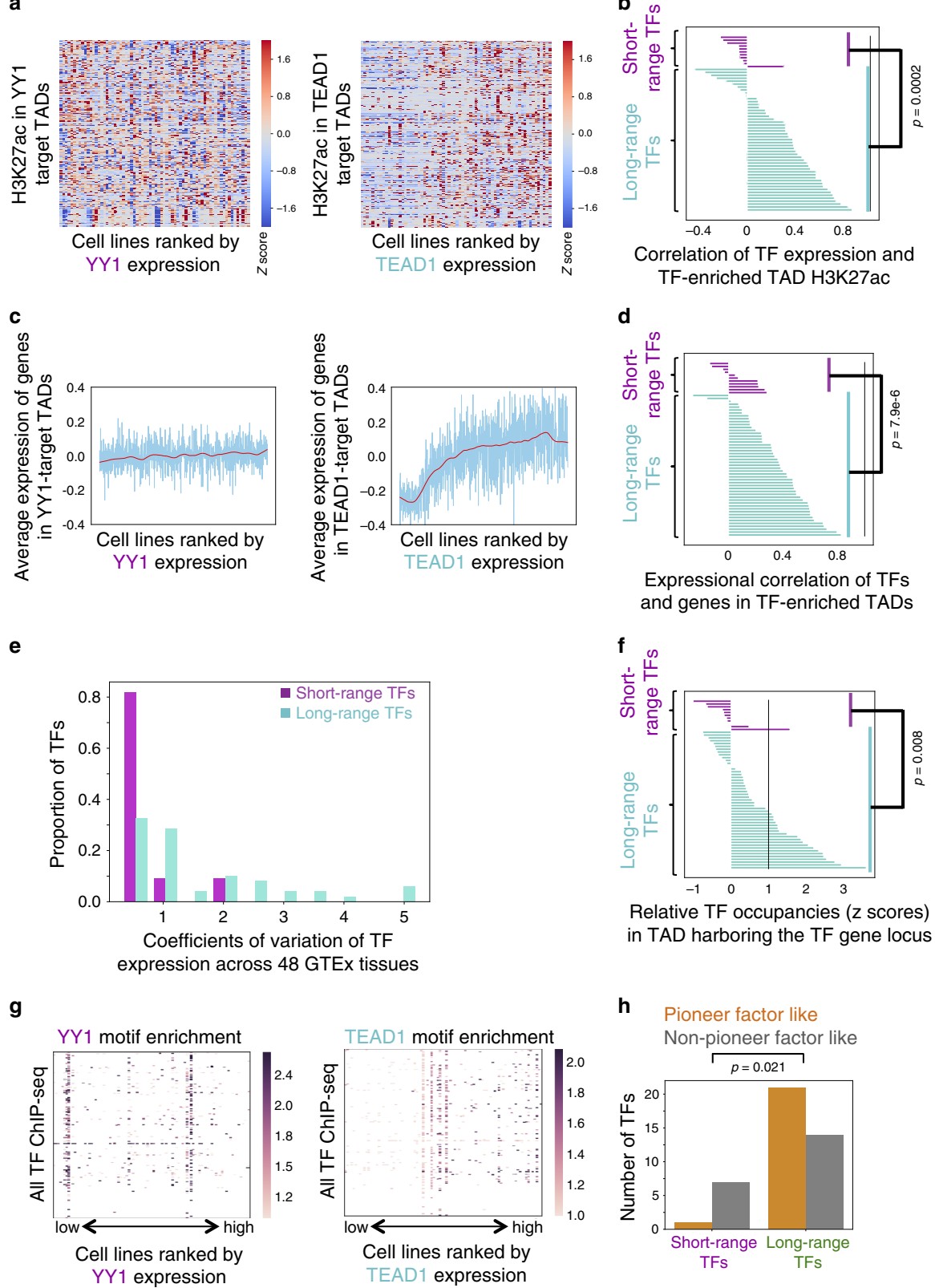

tissues (Fig. 4c), with neural tissues having the largest distance differences. This phenomenon is not due to linkage disequilibrium (LD), as the LD block sizes are similar in B-type and A-type TADs (Supplementary Fig. 4e).

We next examined whether the difference in the regulatory behavior in A-type and B-type TADs is owing to differences in chromatin interaction frequencies within these TAD types. Using Hi-C chromatin interaction data[40], we compared contact frequencies within A-type TADs to those within B-type TADs as a function of genomic distance. Decay rates in Hi-C data are typically quantified using power-law models[11]. Therefore, for a fair comparison, we modified the RP model using the power-law

**Fig. 3 Long-range TFs and short-range TFs have distinct regulatory properties. a** Right: increase of H3K27ac levels in TEAD1 target TADs in cell lines with high TEAD1 expression (*p* value 0.003; two-tailed Pearson correlation). Left: no increase of H3K27ac levels in YY1 target TADs with YY1 expression. x-axis: CCLE cell lines ranked by YY1 (left) or TEAD1 (right) gene expression. *y* axis: YY1 (left) or TEAD1 (right) target TADs. Heatmap colors represent H3K27ac ChIP-seq signal for cell line – TAD pairs. **b** Strong correlation between TF expression and H3K27ac ChIP-seq signal in TF target TADs for long-range TFs but not for short-range ones. (*p* value = 0.0002; two-tailed Student's *t* test). **c** Right: the average expression of genes in TEAD1 target TADs is high in cell lines with high TEAD1 expression. Left: there is no association between YY1 gene expression and gene expression in YY1 target TADs. **d** Strong correlation between TF expression and expression of genes in TF target TADs for long-range TFs but not for short-range ones. (*p* value 7.9e−6; two-tailed Student's *t* test). **e** Tissue-restricted expression of long-range TFs. Compared with short-range TFs, long-range TFs have higher expression coefficients of variation across 48 GTEx tissues. **f** Long-range TFs (green) have more binding sites within the TAD harboring the TF gene itself (*p* value 0.008; two-tailed Student's *t* test). **g** Right: ChIP-seq peaks of TFs besides TEAD1 are more highly enriched with the TEAD1 motif in cell lines with higher TEAD1 expression. Left: no such association is observed in the YY1 case. x axis: CCLE cell lines ranked by YY1 (left) or TEAD1 (right) gene expression. *y* axis: ChIP-seq of TFs besides TEAD1 or YY1. Each cell represents the enrichment of YY1 (left) or TEAD1 (right) motifs in the ChIP-seq peaks of the corresponding TF-cell line pair. **h** Long-range TFs tend to possess pioneer-like properties. The "pioneer-like" property of a TF is defined as the association between the level of that TF's expression and its motif enrichment in the ChIP-seq peaks of other TFs (*p* value 0.02; two-sided Fisher's exact test). Source data are provided as a Source Data file.

equation (methods) and determined the decay constants of long-range TFs to lie in a range between −0.35 and −0.47. The model determined decay rates to be steeper in A-type TADs (−0.4 ~ −0.5) than in B-type TADs (−0.3 ~ −0.45) (Fig. 4d), consistent with what we described above. These decay rates are smaller (in terms of absolute magnitude) than those observed in Hi-C data, which are well beyond −0.5 (Fig. 4e). Although it is interesting to note the differences in A-type and B-type TAD Hi-C decay rates in the 250 kb to 1 Mb range, this range is beyond the regulatory range of TFs observed in our model-based analysis.

**Regulatory distances change with chromatin states.** Our analyses revealed that the binding of long-range TFs in A-type and B-type TADs have different regulatory ranges. However, several genetic properties of TADs contribute to the establishment of their chromatin states and influence their designation as A-type or B-type. To tease out the true chromatin effect from the genetic properties, we next assessed whether different chromatin states in the same TADs in two different cell types, have different impacts on regulatory distances (Fig. 5a). We identified the nuclear factor NF-kappa-B p65 subunit, RELA, for which gene expression data in GTEx and multiple ChIP-seq data sets are available in the lymphoblastoid cell line GM12878 and lung carcinoma cell line A549. We defined TADs with statistically significant differences in H3K27ac ChIP-seq signals between the two cell lines as GM12878-predominant (more H3K27ac in GM12878 than in A549) or A549-predominant TADs. We then used GTEx lymphoblastoid and lung tissue expression data to find the regulatory decay distances of *RELA* in the GM12878-predominant TADs ($\Delta_i^{GM12878}$) and A549-predominant TADs ($\Delta_i^{A549}$), respectively. We observed that the regulatory decay distance estimated using lymphoblastoid gene expression data are shorter in GM12878-predominant TADs ($\Delta_i^{A549} > \Delta_i^{GM12878}$, lymphoblastoid expression). Conversely, the *RELA* decay distance estimated using lung gene expression data is shorter in A549-predominant TADs ($\Delta_i^{GM12878} > \Delta_i^{A549}$, lung expression) (Fig. 5b). This suggests that the regulatory decay distance for a given TF becomes shorter in a given TAD as that TAD becomes more active.

To explore this phenomenon more broadly, we examined the chromatin effects on regulatory decay distance using eQTL data from various tissues and cell types, including brain cortex, stomach, lymphoblastoid, and whole blood. We then identified the tissue-restricted TADs and determined whether the eQTLs within these TADs are closer to their associated genes in the tissues where these TADs are more active. For example, the eQTLs derived from brain cortex tissue tend to be closer to the TSSs of their associated genes in the TADs that are more active in

brain. Likewise, lymphoblastoid eQTLs are closer to the TSSs of the associated genes in the TADs that are more active in lymphoblastoid cells (Fig. 5c). This is not a result of ascertainment bias because such bias would lead to the opposite result: there is greater statistical power to detect weak, distant, eQTL associations for highly expressed genes. In fact, this phenomenon holds true for most pairs of GTEx tissues with available H3K27ac ChIP-seq data we examined (Supplementary Fig. 5). These results support our hypothesis that chromatin states of the TADs, in addition to their genetic properties, influence regulatory decay distances. More specifically, $\Delta_i^*$ becomes shorter in the same TADs when the TADs become active. Further investigation is needed to understand the mechanisms underlying this phenomenon.

**A TAD-wise TF-regulation model helps interpret GWAS.** Genome-wide association studies (GWAS) have determined many trait-associated single-nucleotide polymorphisms (SNPs)[41] in the non-coding regions, but the functional interpretation of these non-coding variants remains challenging. GWAS SNPs are enriched in enhancers and DNase hypersensitive regions[42], suggesting roles in gene regulation, presumably by modulating TF-binding affinities[43]. The implicated TFs were inferred as those with enriched motifs or ChIP-seq peaks significantly overlapping causal SNPs, which were fine mapped using statistical evidence or through functional annotation[44]. Such approaches often cannot produce satisfactory results for the following reasons: (1) limited numbers of trait-associated SNPs; (2) sparse TF ChIP-seq peaks across the genome; and (3) differential TF-binding densities in different genomic regions. To illustrate the last point, consider two TFs, TF X and TF Y, each with a total of 10 ChIP-seq peaks genome-wide. A B-type TAD harbors one TF X and three TF Y ChIP-seq peaks, and an A-type TAD harbors nine TF X and seven TF Y peaks. Although the total numbers of peaks are the same for TF X and TF Y, TF Y would play a dominant role in the B-type TAD, whereas playing a similar role as TF X in the A-type TAD. In other words, it is important to consider the density of a particular TF's binding relative to other TFs. Analysis of sparse trait-associated SNPs and sparse TF ChIP-seq peaks can be facilitated by relaxing the requirement for ChIP-seq peaks to overlap SNPs. The trait-associated SNPs in GWAS studies are after all only proxies for causal SNPs and the TADs harboring these causal variants can be well estimated. The same principle applies to TF ChIP-seq: although not all TF-binding sites are necessarily identified in ChIP-seq, the peaks that are observed may well approximate the relative abundances of TF binding within TADs. Therefore, we can infer implicated TFs by

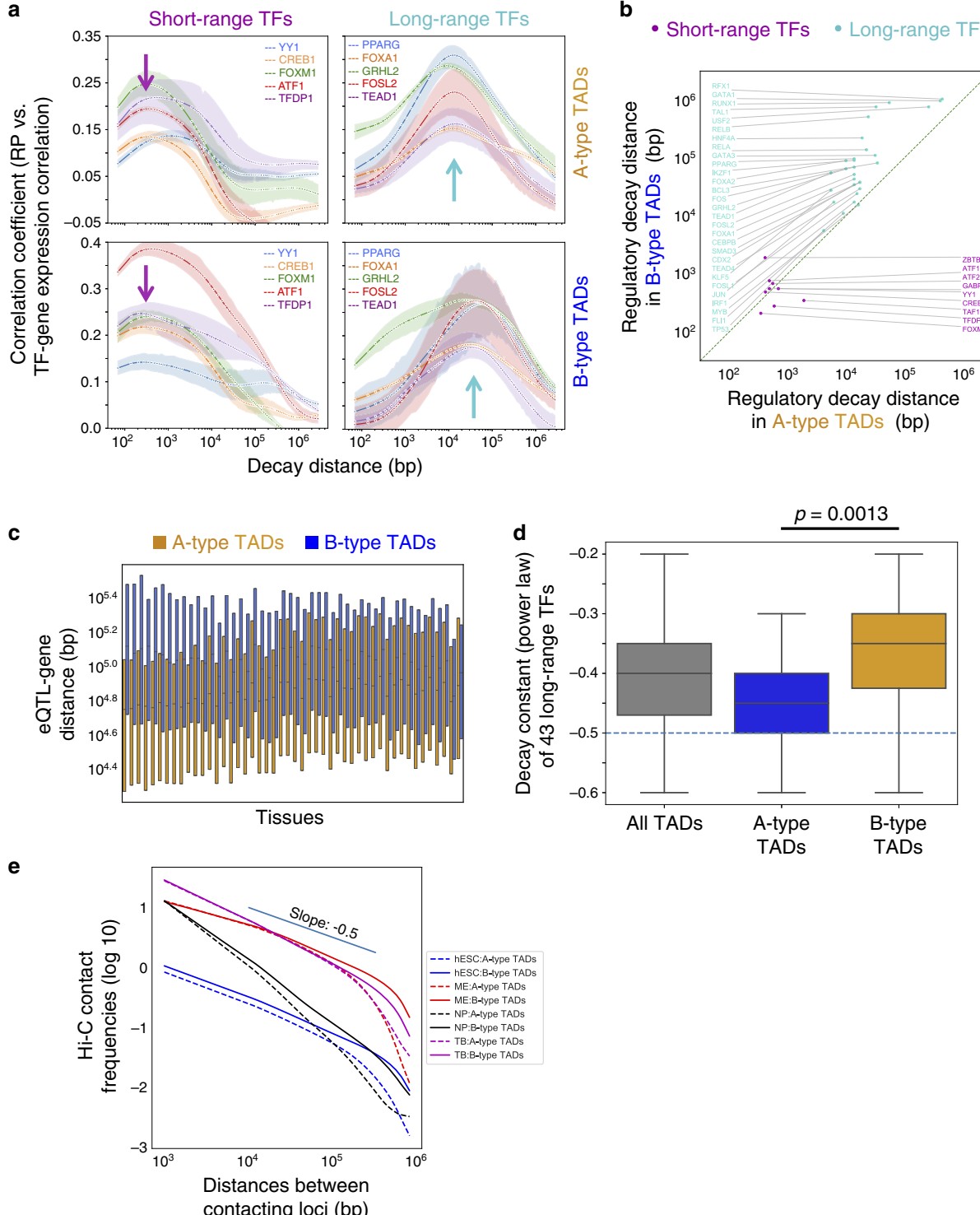

**Fig. 4 Long-range TFs have longer regulatory decay distances in B-type TADs. a–b** Long-range TFs have longer regulatory decay distances in B-type TADs than in A-type TADs. **c** In 47 (out of 48) GTEx tissues, eQTL-TSS distances are longer in B-type (blue) TADs than those in A-type (red) TADs. Tissues were sorted using differential eQTL-TSS distances between A-type and B-type TADs. The box plot extends from the lower to the upper quartile values of the data, with a line at the median. **d** The power-law decay rates of 49 long-range TFs measured in A-type TADs (−0.4 ~ −0.5) are significantly larger (in terms of absolute magnitude) than those measured in B-type TADs (−0.3 ~ −0.45), (two-sided Student's *t* test *p* value = 0.0013). The box plot extends from the lower to the upper quartile values of the data, with a line at the median. The whiskers extend from the box to show the range of the data. **e** In human embryonic stem cells (hESC), mesendoderm (ME), neural progenitor (NP), and trophoblast-like (TB) Hi-C data, the average contact frequency between two loci decreases as the distances between the loci increases, following a power-law relationship. In the range of TAD sizes (median size 680 kb), the decay rates are more negative than −0.5. Source data are provided as a Source Data file.

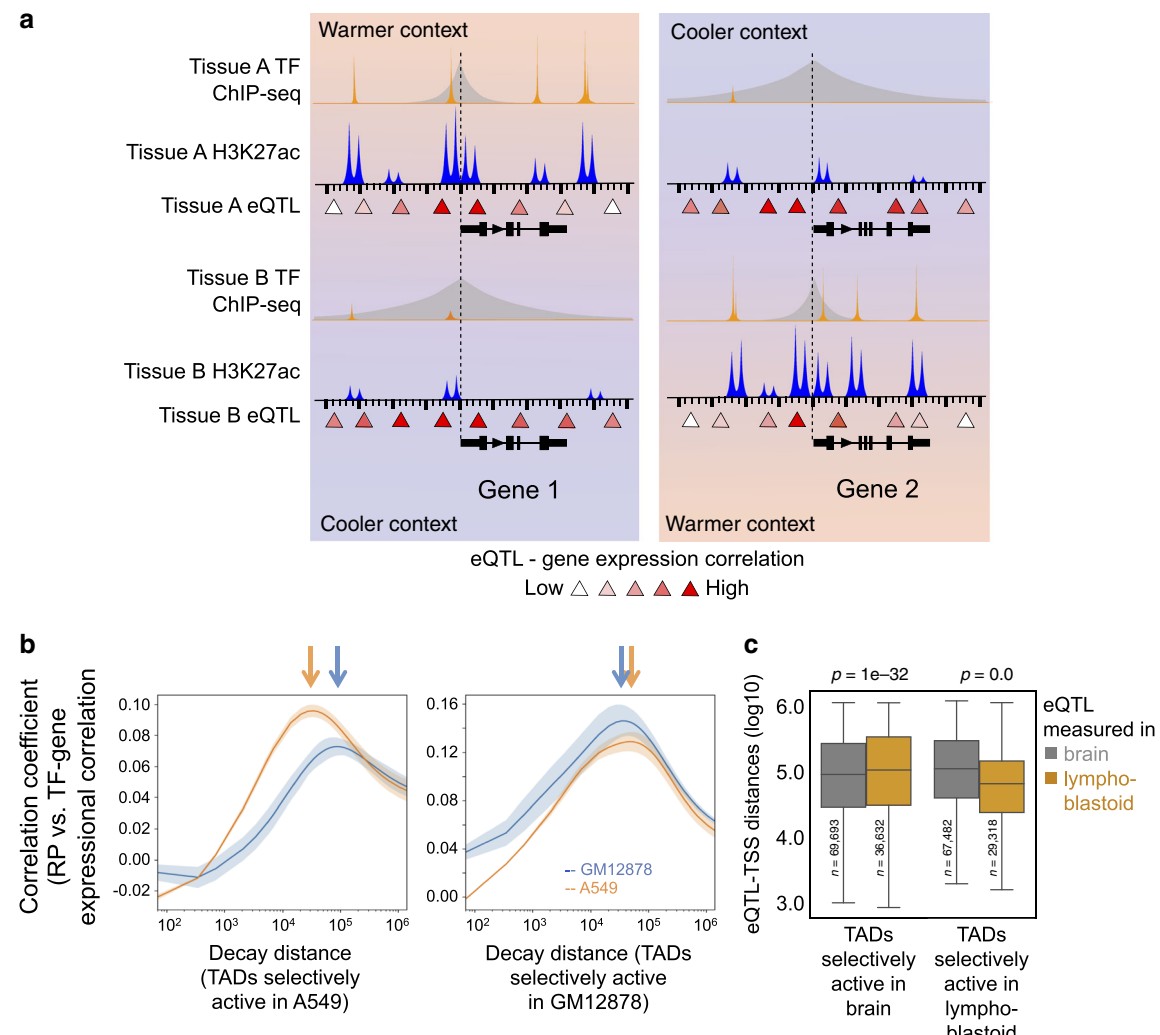

**Fig. 5 Regulatory decay distances change with TAD activity and chromatin state. a** Schematic illustration of TF regulatory decay and eQTL-TSS distances decreasing with increasing TAD activity. **b** The regulatory decay distance of RELA gets shorter as TADs become more active. RELA is expressed in both lymphoblastoid and lung, which possess distinct distributions of active TADs. In the TADs that are more active in lung than in lymphoblastoid cells, the lung-specific RELA regulatory distance is shorter than the lymphoblastoid-specific RELA regulatory distance (left). On the other hand, in the TADs that are more active in lymphoblastoid cells than in lung, the lymphoblastoid-specific RELA regulatory distance is shorter than the lung-specific RELA regulatory distance (right). The lung-specific RELA regulatory decay distance is estimated using RELA ChIP-seq in A549 lung cells and GTEx lung expression data, and the lymphoblastoid-specific RELA regulatory decay distance is estimated using RELA ChIP-seq in GM12878 and GTEx lymphoblastoid expression data. **c** GTEx eQTL-TSS distances are shorter in the TADs that are more active in the tissues in which the eQTLs are measured. As in **b**, the distribution of GTEX eQTL-TSS distances measured in brain (gray) or lymphoblastoid (orange) were compared in TADs that are more active in brain than in lymphoblastoid cells and vice versa. Left: TADs more active in brain than in lymphoblastoid. Right: TADs more active in lymphoblastoid than in brain. The different log-transformed eQTL-TSS distances in individual groups of TADs were compared using the two-sided Student's *t* test. The box plot extends from the lower to the upper quartile values of the data, with a line at the median. The whiskers extend from the box to show the range of the data. Source data are provided as a Source Data file.

overlapping SNPs with TADs, characterized by different relative TF abundances (methods). We inferred the implicated TFs using fine mapped SNPs[44], and found our approach to outperform SNP-TF ChIP-seq peaks overlaps in ranking known diseases-associated TFs[45] (Fig. 6a).

Our approach can also be understood as associating the non-coding SNPs that are likely to affect genes with the TAD-dominant long-range TFs that are likely to regulate these same genes (Fig. 2c–h). Applying this method to infer the possible role or long-range TFs in diseases with >20 annotated associated SNPs in the GWAS catalog[41], we identified clusters of TFs associated with distinct groups of diseases, such as the previously known association of *RUNX1* with autoimmune[46] diseases, *KLF5* with breast cancer[47], and *CEBPA* as well as *HNF4A* with lipid

metabolism[48] (Fig. 6b). This analysis also revealed co-occurring diseases that have been reported in epidemiological studies, such as autoimmune diseases and chronic lymphocytic leukemia[49], as well as type-2 diabetes and breast cancer[50]. These findings demonstrate how a TAD-wise analysis of TF relative binding enrichment could help prioritize long-range TFs germane to traits of interest in GWAS studies.

## Discussion
Despite intensive scientific investigation into the role of TFs in regulating metazoan gene expression, the mechanisms by which TFs regulate specific genes are still not well understood. In this study, we quantitatively modeled the ranges of genomic distance over which TFs regulate genes in different genomic and

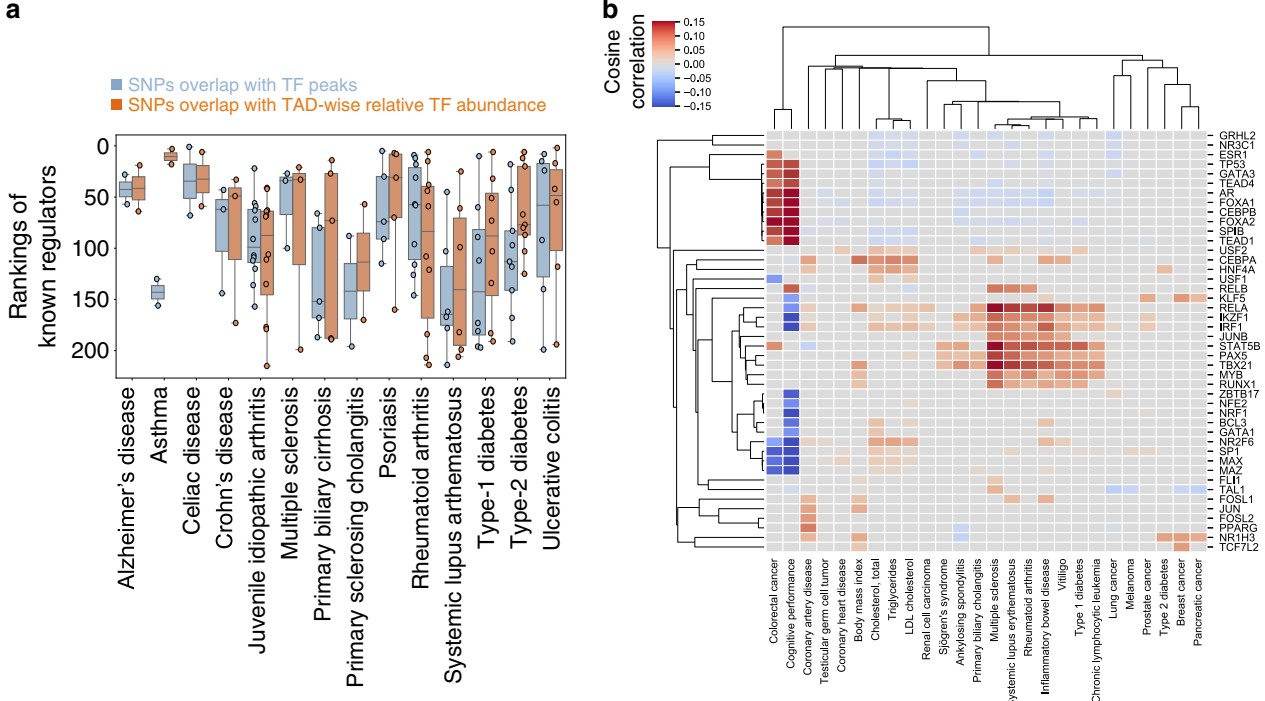

**Fig. 6 TAD-wise analysis of non-coding GWAS hits prioritizes relevant TFs. a** Evaluation of the inference of GWAS-SNP relevant TFs on the basis of rankings of known disease-associated TFs. Blue boxes are known disease-associated TFs ranked by overlapping SNPs with TF ChIP-seq peaks, whereas orange boxes are known disease-associated TFs ranked by overlapping SNPs with TADs characterized by different relative TAD-wise TF abundances. The later approach can be approximated as cosine similarities between relative TAD-wise TF occupancies and numbers of disease-associated SNPs in TADs. The two-sided Wilcoxon signed-rank test $p$ value of the paired median TF rankings is 0.005 ($n = 14$). The box plot extends from the lower to the upper quartile values of the data, with a line at the median. The whiskers extend from the box to show the range of the data. **b** The heatmap shows the long-range TFs implicated from GWAS SNPs. The x axis represents traits in the GWAS Catalog, especially autoimmune disease, cancers, and metabolic syndromes, and the y axis are inferred long-range TFs. Each entry is the color-coded cosine similarity between relative TF occupancies in TADs and numbers of trait-associated SNPs in TADs. Red squares or blue squares are trait-TF pairs with significantly ($p$ value < 0.001 using one-sided permutation considering multiple hypothesis test) positive or negative cosine similarities, respectively. TFs and traits were clustered using hierarchical clustering. Source data are provided as a Source Data file.

chromatin contexts. We systematically examined 3406 TF ChIP-seq, 1545 H3K27ac histone modification ChIP-seq, gene expression from 1037 CCLE cancer cell lines and 48 GTEx tissues, and further analyzed eQTL and GWAS catalog data.

Our analyses revealed two distinct classes of TF with different ranges of regulatory influence, chromatin-binding preferences, pioneer-like properties, and auto-regulatory behaviors. Although the precise mechanisms that underlie these regulatory classes remain unclear, it is clear that different TFs have different regulatory genomic ranges of influence. Regarding the TF properties explored in this study, classifying TFs into two groups is sufficient to contrast their differences. Our results suggest that the binding sites of short-range TFs that are located far away from gene TSSs, which usually account for the majority of binding sites, are unlikely to influence gene expression. Transcription is known to involve several distinct stages including recruitment of RNA polymerase II (PolII) to the promoter and release of the paused early elongation complex into productive elongation[51]. We speculate that short-range TFs might participate primarily in the PolII recruitment and initiation stages, whereas the long-range TFs might primarily influence elongation. A division in labor for transcriptional regulatory proteins has been reported for a small number of TFs and cofactors. For example, Bromodomain and ExtraTerminal proteins, mediator and the p-TEFb complexes are known to be important for productive transcriptional elongation[51], and the TF *Sp1* has a known role in PolII pre-initiation complex recruitment[52,53]. In addition, *Sp1* and *CTF* activation

domains have been found to stimulate initiation, whereas VP16, p53, and E2F1 stimulate both initiation and elongation[54,55]. Our results suggest such a division of labor applies to many DNA-binding TFs. Recent observations that promoters can act like enhancers for other genes[56] might depend on the recruitment of long-range TFs to enhancer-like promoters.

The genome is compartmentalized into TADs; some TADs are "B-type" with low levels of activity in most cell types, whereas others are "A-type" and highly active in most cell types. Known cases of genes being regulated by enhancers over extremely long genomic distances mostly occur in B-type TADs, prompting us to examine whether these observations were part of a general trend. Our analysis of TF ChIP-seq and eQTL data shows that, in general, regulatory decay distances are longer in B-type TADs than in A-type ones. By measuring the regulatory distance in the same TADs under different chromatin states, we found that regulatory decay distances become shorter when chromatin becomes more active. As it is widely believed that enhancers regulate target genes through direct physical loop formation, we explored Hi-C data to determine whether the Hi-C interactions in A-type and B-type TADs could explain our results, but they did not. It is possible that current interaction–detection technologies or data processing techniques do not capture all regulatory interactions on the relevant time and length scales or that non-looping mechanisms may mediate certain enhancer functions. The discordance between the interaction data and our decay distance analysis corresponds with recent high-resolution microscopy experiments on the regulation

of *SOX2*[57] and *SHH*[58], which are regulated by certain distal enhancers without direct physical enhancer–promoter interaction. In addition, a recent study in Drosophila showed that disrupting long-range inter-TAD loops does not alter expression for the majority of genes[59].

Non-looping regulatory mechanisms may involve thermo-dynamically induced phase separation phenomena[60]. The compartments that have been observed in Hi-C data suggest that heterochromatin and euchromatin regions of the genome are spatially segregated[5] and may form different nuclear structures such as lamin-associated domains[61], nucleoli, nuclear speckles, PML bodies, and Cajal bodies[62]. In addition, altering the nuclear position of a locus has been observed to modify the expression of nearby genes[63]. Our analysis raises the possibility that TF-induced disruption of heterochromatin, by disruption of transcriptionally repressive zones or by nucleation of transcriptionally active ones, can lead to alterations in the regulatory microenvironment of genes and account for the chromatin state-dependent regulatory decay distances. Recent experimental work using a CRISPR-Cas9 based technology to induce phase-separated chromatin droplet formation at targeted genomic loci has shown that droplet formation in heterochromatin regions can cause large disruptions of heterochromatin domains[64]. The coordinated binding of pioneer factors to heterochromatin rich TADs could seed droplet nucleation points[64] and the enhancer activity of these binding sites might be related to the release of a gene from a repressive environment. An alternative mechanism is facilitated tracking, in which enhancer-bound protein complexes move toward the promoter in a progressive, unidirectional fashion, whereas possibly remaining bound to the enhancer[65]. In principle, the presence of enhancers or promoters between the tracking enhancer and its target promoter may impede the tracking progress and would explain the shorter regulatory distances in A-type domains.

Our modeling of regulatory decay distances is based on large-scale TF cistrome data and gene expression cohorts of various cell lineages. Although the Cistrome DB contains most publicly available TF cistromes, and the CCLE and GTEx expression cohorts include many different cell and tissue types, we are far from covering all the TFs in all cellular contexts. Nonetheless, the high degree of consistency between analyses using large amounts of orthogonal data provides solid support that the trends we observed are likely to reflect the general behavior of most TFs. These findings may help the interpretation of GWAS SNPs: significant SNPs in B-type TADs can impact the expression of distant genes in the same TAD; significant SNPs in A-type TADs, on the other hand, are less likely to be associated with (>100 kb) distant genes. Some TADs are densely bound by certain long-range TFs, and the chromatin states of these TADs are dominated by these TFs. These TADs, which are "targets" of TFs, are predominantly of the B-type. Genes located in these target TADs usually have tissue-specific functions and the target TADs often harbor the TF gene itself, indicating TAD-wise autoregulatory control of cell lineage specification. Integration of the heterogeneous distribution of long-range TFs and GWAS SNPs can better prioritize the TFs implicated in the manifested phenotype than the conventional SNP-TF ChIP-seq overlap approach. A hierarchical probabilistic framework in which relative TAD-wise TF abundance combined with local TF information may further improve the inference. Further work are needed to understand the mechanistic basis of these TF-specific and context-dependent gene regulatory effects.

## Methods

**TAD annotation**. The TAD annotations were downloaded from http://chromosome.sdsc.edu/mouse/hi-c/download.html, and their coordinates were converted from hg18 to hg38 using liftOver software from UCSC: http://hgdownload.cse.ucsc.edu/goldenPath/hg18/liftOver/.

**TF ChIP-seq data processing**. The raw sequence data of TF ChIP-seq were downloaded from Gene Expression Omnibus and processed through standard workflow of ChiLin[66], consisting of quality control and peak calling using MACS. For fair comparisons of TF occupancy distributions between samples, those TF ChIP-seq samples with <20,000 peaks were discarded, and for other TF ChIP-seq samples only the top 20,000 peaks based on peak intensities were included for downstream analysis.

**Identification of TF enriched TADs**. For each TF ChIP-seq sample, we calculated the TF-binding density in each TAD (number of peaks/kb). Noticing that for all TFs this density is higher in the A-type TADs than the B-type TADs, we examined whether some TADs are more densely bound by certain TFs relative to others. To achieve this, for each TAD we calculated the mean and standard deviation of the TF-binding density in the TAD across all TFs. We then applied a z score transformation to all TF ChIP-seq samples, and calculated the z score of a given TF to get the TAD-specific relative occupancies of the given TF compared with other TFs. For example, a TF $i$ with a high z score in a given TAD $j$, indicates that TAD $j$ is more densely bound by TF $i$ in comparison with other TFs. For each TF, we define TF target TADs as those TADs with z scores higher than 1.

**RP model**. In our model of transcription regulation by a given TF $i$, we assume that multiple TF $i$ binding sites contribute additively to the regulation of a gene $j$. In this model, we modeled the effect of a single ChIP-seq peak $k$ of TF $i$ on gene $j$ with an influence function that decays exponentially with the genomic distance between the TSS of gene $j$ and peak $k$, $x_{i,j,k}$. The decay distance ($\Delta$) defines the half-life of the exponential decay function. The RP, $R_{i,j}(\Delta)$, defines the total regulatory effect of TF $i$ on gene $j$ by summing all binding sites of TF $i$ within TAD of the gene $j$ (Eq. 1).

$$R_{i,j}(\Delta) = \sum_{\text{peak } k \text{ of TF } i} 2^{\frac{-x_{i,j,k}}{\Delta}} \tag{1}$$

This RP model can be also built using power-law functions:

$$R_{i,j}(\lambda) = \sum_{\text{peak } k \text{ of TF } i} x_{i,j,k}^{\lambda}$$

Where $\lambda$ is power-law decay rate.

**Inferring TF regulatory ranges using TF perturbations**. Assuming that the DE genes upon TF perturbation would comprise TF direct target genes, we hypothesize that the RP, $R_{i,j}(\Delta)$, with correct regulatory decay distance can best separate DE genes from other genes. In TF perturbation related gene expression data matched to TF ChIP-seq data, we calculated the $R_{i,j}(\Delta)$ for each gene using Eq. (1) and derived DE genes (absolute log₂ fold change > 1 and p value < 0.01). We then used Kolmogorov–Smirnov test to measure how $\Delta$ separates DE genes from other genes. The regulatory decay distance $(\Delta_i^*)$ was then defined as the $\Delta$ that gave rise to the most significant KS test value.

**Inferring TF regulatory ranges using gene expression cohorts**. The method for inferring the regulatory decay distance of a TF $i$ is based on the relationship between TF $i$ gene expression, the gene expression of other genes, and the ChIP-seq peaks of TF $i$. In step 1, we calculate a statistic (the Pearson correlation coefficient) that summarizes the association between the expression of TF $i$ and the expression of the other genes. In step 2, the summary statistic from step 1 is combined with TF ChIP-seq data in a model of cis-regulatory effect that uses RPs to estimate the regulatory decay distance. In step 2, a range of values for the regulatory decay distance parameter $\Delta_i$ for TF $i$ is evaluated to determine which value of $\Delta_i$ maximizes the association between the TF $i$ ChIP-seq data and the gene expression correlation of TF $i$ gene with other genes.

Step 1: Association between the gene expression of a TF and other genes. We make the following two assumptions about the relationship between a TF's gene expression and the expression of the genes it regulates: (1) TFs are regulators of gene expression, and (2) TFs do not influence gene expression unless they are expressed. We approximate the relationship between the gene expression of TF $i$ and another gene $j$ in sample $k$ using the linear relationship:

$$E_{j,k} = \alpha + \gamma_{i,j} E_{i,k} + \epsilon_j \tag{2}$$

where $E_{i,k}$ and $E_{j,k}$ are the standardized expression levels of TF $i$ and gene $j$ in sample $k$ (i.e., $E_i \sim N(0,1)$ and $E_j \sim N(0,1)$). $\gamma_{i,j}$ summarizes the transcriptional regulatory effect TF $i$ might have on gene $j$, and $\epsilon_j \sim N(0, \sigma^2)$. Below we show that, under these assumptions, the Pearson correlation coefficient ($\rho_{i,j}^{\text{expr}}$) between the gene expression of TF $i$ ($E_i$) and another gene $j$ ($E_j$) is the maximum likelihood estimator of the regulatory effect parameter, $\gamma_{i,j}$.

Given expression data $(E_{i,1}, E_{j,1})$ $(E_{i,2}, E_{j,2}) \ldots$ $(E_{i,n}, E_{j,n})$, the conditional probability distribution of the data under Eq. (2) is:

$$\prod_{k=1}^{n} p\left(E_{j,k}|E_{i,k}; \alpha, \gamma_{i,j}, \sigma^2\right) = \prod_{k=1}^{n} \frac{1}{\sqrt{2\pi\sigma^2}} e^{-\frac{(E_{j,k} - (\alpha + \gamma_{i,j}E_{i,k}))^2}{2\sigma^2}}$$

The log-likelihood of the parameters based on the observed expression data is:

$$l\left(\alpha, \gamma_{i,j}, \sigma^2\right) = \sum_{k=1}^{n} \log p\left(E_{j,k}|E_{i,k}; \alpha, \gamma_{i,j}, \sigma^2\right)$$

$$= -\frac{n}{2}\log 2\pi - n\log\sigma - \frac{1}{2\sigma^2}\sum_{k=1}^{n}(E_{j,k} - (\alpha + \gamma_{i,j}E_{i,k}))^2$$

$\hat{\gamma}_{i,j}$, the maximum likelihood estimator of $\gamma_{i,j}$, can be derived using $\frac{\partial l(\alpha, \gamma_{i,j}, \sigma^2)}{\partial \gamma_{i,j}} = 0$:

$$\hat{\gamma}_{i,j} = \frac{\sum_{k=1}^{n}\left(E_{j,k} - \overline{E_j}\right) \cdot \left(E_{i,k} - \overline{E_i}\right)}{\sum_{k=1}^{n}\left(E_{i,k} - \overline{E_i}\right)^2} = \frac{C_{ij}}{S_i^2}$$

where $C_{ij}$ is the covariance between $E_i$, and $E_j$. As $E_{i,k}$ and $E_{j,k}$ are standardized expression levels of TF $i$ and gene $j$, and thus $S_i^2 = S_j^2 = 1$.

$$\begin{aligned}
\hat{\gamma}_{i,j} &= \frac{C_{ij}}{S_i^2} \\
&= \frac{S_j}{S_i} \cdot \frac{C_{ij}}{S_i \cdot S_j} (S_i = S_j) \\
&= \frac{C_{ij}}{S_i \cdot S_j} \text{ (Definition of Pearson correlation coefficient)} \\
&= \rho_{i,j}^{\text{expr}}
\end{aligned} \tag{3}$$

Where $\rho_{i,j}^{\text{expr}}$ is the Pearson correlation coefficient between $E_i$ and $E_j$.

Step 2: Association between gene expression correlation and TF ChIP-seq data. We assume that, for some value $\Delta_i$, the ChIP-seq peak derived RP model, $R_{i,j}(\Delta_i)$, can approximate $\hat{\gamma}_{i,j}$, the gene expression based estimator of the possible transcriptional regulatory effect of TF $i$ on gene $j$:

$$\hat{\gamma}_{ij} = f\left(R_{i,j}\left(\Delta_i^*\right)\right) + \epsilon_{ij} \tag{4}$$

$f(\cdot)$ can be a linear or nonlinear function, and we conduct the analysis using both linear and nonlinear model.

**Linear model (for Step 2)**. $\hat{\gamma}_{i,j}$, the cis-regulatory effect of TF $i$ on gene $j$, can be approximated as:

$$\hat{\gamma}_{ij} = \alpha_i + \beta_i R_{i,j}\left(\Delta_i^*\right) + \epsilon_{ij} \tag{5}$$

Where $R_{i,j}(\Delta_i^*)$ is the RP model of cis-regulatory effect of TF $i$ on gene $j$ parameterized by TF-specific regulatory decay distance $\Delta_i^*$, defined as (1), and $\epsilon_{ij} \sim N(0, \sigma_i^2)$.

The parameters in Eq. (5) can be inferred using maximum likelihood estimations:

$$\hat{\alpha}_i, \hat{\beta}_i, \widehat{\Delta_i^*}, \widehat{\sigma_i^2} = \text{argmax}\left(-\frac{n}{2}\ln(2\pi) - \frac{n}{2}\ln(\sigma_i^2) - \frac{1}{2\sigma_i^2}\sum_{j=1}^{n}\left(\hat{\gamma}_{i,j} - \alpha_i - \beta_i \cdot R_{i,j}\left(\Delta_i^*\right)\right)^2\right) \tag{6}$$

Where $n$ is the number of genes. To get the maximum likelihood estimator $\hat{\alpha}_i, \hat{\beta}_i, \widehat{\Delta_i^*}, \widehat{\sigma_i^2}$, we first fixed $\Delta_i^*$, and $\hat{\alpha}_i, \hat{\beta}_i, \widehat{\sigma_i^2}$ are:

$$\hat{\beta}_i = \frac{\sum_{j=1}^{n}\left(\hat{\gamma}_{i,j} - \overline{\hat{\gamma}_{i,j}}\right)\left(R_{i,j}(\Delta_i^*) - \overline{R_{i,j}(\Delta_i^*)}\right)}{\sum_{j=1}^{n}\left(R_{i,j}(\Delta_i^*) - \overline{R_{i,j}(\Delta_i^*)}\right)^2}$$

$$\hat{\alpha}_i = \overline{\hat{\gamma}_{i,j}} - \hat{\beta}_i \cdot \overline{R_{i,j}(\Delta_i^*)}$$

$$\widehat{\sigma_i^2} = \frac{1}{n}\sum_{j=1}^{n}\left(\hat{\gamma}_{i,j} - \hat{\alpha}_i - \hat{\beta}_i \cdot R_{i,j}\left(\Delta_i^*\right)\right)^2$$

Then with fixed $\hat{\alpha}_i, \hat{\beta}_i, \widehat{\sigma_i^2}$,

$$\widehat{\Delta_i^*} = \text{argmax}\left(-\frac{n}{2}\ln\left(\widehat{\sigma_i^2}\right)\right) = \text{argmin}\left(\widehat{\sigma_i^2}\right) = \text{argmax}(\rho^2) \tag{7}$$

Where $\rho$, the Pearson correlation coefficient between $\hat{\gamma}_{i\cdot}$ and $R_{i\cdot}(\Delta_i^*)$, is given by:

$$\rho_i^{\text{expr,RP}(\Delta_i^*)} \equiv \rho_i^{\hat{\gamma}_{i\cdot}, R_{i\cdot}(\Delta_i^*)} = \frac{\sum_{j=1}^{n}\left(\hat{\gamma}_{i,j} - \overline{\hat{\gamma}_{i,j}}\right)\left(R_{ij}(\Delta_i^*) - \overline{R_{i,j}(\Delta_i^*)}\right)}{\sqrt{\sum_{j=1}^{n}\left(\hat{\gamma}_{i,j} - \overline{\hat{\gamma}_{i,j}}\right)^2}\sqrt{\sum_{j=1}^{n}\left(R_{i,j}(\Delta_i^*) - \overline{R_{i,j}(\Delta_i^*)}\right)^2}} \tag{8}$$

Therefore, to approximate $\widehat{\Delta_i^*}$, the maximum likelihood estimator of $\Delta_i^*$, we tested a range of different $\Delta$ values from 100 bp to 4000 kb, and determined $\widehat{\Delta_i^*}$, as the one that maximizes Eq. (7).

**Nonlinear model (for Step 2)**. In an alternative method we measure the association between the gene expression association, $\hat{\gamma}_{i\cdot}$, and the ChIP-seq based RP, $R_{i,j}(\Delta)$, using "distance correlation" (dCor), a measure of association that, unlike the Pearson correlation, is independent of the underlying variable distributions[67]. Systematically varying $\Delta$, regarding $R_{i,j}(\Delta)$ with different $\Delta$ as different models, we perform model selection using goodness of fit tests:

$$\Delta_i^* = \text{argmax}_{\Delta}\left(\text{dCor}\left(\hat{\gamma}_{ij}, R_{i,j}(\Delta)\right)\right)$$

**Empirical assessment of assumptions**. The purpose of this model is to use gene expression to quantify the possible regulatory relationships between a TF and other genes, based on an analysis of gene expression and TF ChIP-seq data. In the empirical evaluation of Step 2 (Supp. Fig. 2a) we find that $\hat{\gamma}_{ij}$ and $R_{i,j}(\Delta_i^*)$ are indeed significantly correlated, suggesting that $\Delta_i^*$ inferred using a linear relation in Eq. (5) captures the association between $\hat{\gamma}_{ij}$ and $R_{i,j}(\Delta)$. For all the TFs used in this study, the $p$ values were found to lie in a range between $10^{-200}$ (< machine precision $\sim 10^{-16}$) and $10^{-8}$. Supp. Fig. 2d, e show that the inferred regulatory distances derived on the basis of linear and distance correlation based approaches are similar.

**Procedure for inferring regulatory ranges of TFs**. Based on Eq. (7), we can estimate TF $i$-specific $\Delta_i^*$ by integrating TF ChIP-seq from Cistrome and gene expression data from CCLE and other gene expression cohorts. The detailed steps are as follows:

1. We modeled $\hat{\gamma}_{i,j}$, the maximum likelihood estimator of transcriptional regulatory effect of TF $i$ on gene $j$, with $R_{ij}(\Delta_i^*)$ using Eq. (5). $R_{i,j}(\Delta_i^*)$ was calculated based on TF ChIP-seq data with Eq. (1).
2. In Eq. (3), $\hat{\gamma}_{i,j}$ equals to the Pearson correlation coefficient, $\rho_{i,j}^{\text{expr}}$, between the expression of TF $i$ and gene $j$, using normalized gene expression data. The expression data can be CCLE, GTEx, or other gene expression cohorts.
   a. CCLE: we downloaded CCLE gene expression data and normalized the log-transformed RPKM values in different cell lines using quantile normalization, so that the distributions of gene expression values were the same in each cell line. Data were then normalized gene-wise by mean centering.
   b. GTEx: to infer tissue-specific $\Delta_i^*$, we matched the ChIP-seq cell line to GTEx tissue type. We downloaded normalized GTEx gene expression data, and derived tissue-specific Pearson correlation $\rho_{i,j}^{\text{expr}}|$tissue.
3. As Eq. (6), $\widehat{\Delta_i^*}$ can be inferred as the $\Delta$ that gives rise to the maximum $\rho_i^{\text{expr,RP}(\Delta)}$ as Eq. (7). Therefore, we repeated the above step 1–2 with $\Delta$ ranging from 100 bp to 4000 kb, and defined $\Delta_i^*$ as:

$$\widehat{\Delta_i^*} = \underset{\Delta}{\text{argmax}}\left(\text{abs}\left(\rho_i^{\text{expr,RP}(\Delta)}\right)\right)$$

Specifically, we derived the $\widehat{\Delta_i^*}$ using the following steps:

   a. We defined qualified TF ChIP-seq samples if there is any $\Delta$ giving rise to $\rho_i^{\text{expr,R}(\Delta)}$ greater than 0.1.
   b. If there were more than two qualified TF $i$ ChIP-seq samples, we averaged the $\rho_i^{\text{expr,R}(\Delta)}$ for each $\Delta$. If the maximum average correlation is larger than 0.1, we chose the $\Delta$ that gives rise to the maximum average correlation coefficient as the $\Delta_i^*$.
   c. For GTEx, we matched the ChIP-seq cell line to GTEx tissue type and derived tissue-specific regulatory decay distance, $\Delta_i^*|$tissue as step 3a–b.

**Modeling dynamic regulatory distances of TFs**. For the investigation of the regulatory decay distance dependence on chromatin context, we proceeded as follows:

1. To infer TAD type-specific (e.g., B-type TADs or A-type TADs) regulatory decay distances, in step 3 of "Inferring regulatory decay distances of transcriptional factors using Cistrome TF ChIP-seq data and CCLE or GTEx gene expression data" we included only genes located within the designated

TAD type (e.g., B-type TADs or A-type TADs). Other steps remained the same.

2. To investigate whether the regulatory decay distances depend on chromatin status in addition to genomic features, we focused on TADs that have distinct activity status levels in different lineages. Take lung and lymphoblastoid tissues, for example. We first computed the lung H3K27ac level for each TAD by averaging the TAD H3K27ac mean values over all H3K27ac ChIP-seq samples associated with the lung cluster. The lymphoblastoid H3K27ac TAD levels were calculated in a similar way. We defined the lung-specific TAD clusters as those with significantly ($p$ value $< 1e-10$) higher H3K27ac levels in lung compared with H3K27ac ChIP-seq produced from lymphoblastoid. The lymphoblastoid-specific TAD clusters were defined in a similar way. For the RELA analysis we used GTEx lung expression data and RELA ChIP-seq samples in A549 to compute the lung-specific regulatory decay distance. RELA ChIP-seq in lymphoblastoid and GTEx lymphoblastoid expression data was used to compute the lymphoblastoid-specific regulatory decay distance.

**H3K27ac clustering**. The raw sequence data of H3K27ac ChIP-seq were downloaded from Gene Expression Omnibus and processed through standard workflow of Chilin[66] with peak calling using MACS and the processed data are accessible through the Cistrome Data Browser. We calculated the average H3K27ac occupancies in each TAD using bigWigAverageOverBed, and further normalized each sample using $z$ transformation. We then clustered the H3K27ac CHIP-seq samples and TAD as 10 clusters, both using hierarchical clustering. We defined the TAD cluster with the weakest and strongest H3K27ac signals as B-type TADs and A-type TADs, respectively.

**Pioneer-like factor identification**. Pioneer factors can access closed chromatin and facilitate the binding of other TFs to the accessible genomic loci. To probe whether a TF such as TEAD1, for example, possesses pioneer-like factor properties, we investigated whether there is a gain in TEAD1 motif enrichment in peaks of non-TEAD1 TF ChIP-seq samples in cell lines with high levels of TEAD1 expression. We processed as following steps:

1. Calculate the TEAD1 motif enrichment in peaks of all non-TEAD1 TF ChIP-seq samples using Homer[68]. TEAD1 motif enrichment represents the ratio of the proportion of non-TEAD1 ChIP-seq peaks with the TEAD1 motif to the proportion of background sequences with the TEAD1 motif.
2. Average the TEAD1 motif enrichment for redundant TF cell pairs.
3. Define the median TEAD1 enrichment across all non-TEAD1 TF ChIP-seq samples in each cell line $i$ as $y_i$ and define $x_i$ as the level of TEAD1 gene expression in that cell line. Defining $\bar{y}$ as the mean of $y_i$ across all cell lines, we further derived $y_i^*$ as follows:

$$y_i^* = 1(y_i \geq \bar{y})$$

4. Perform logistic regression $y_i^* \sim x_i$, and define TEAD1 as "pioneer-factor like" if the logistic regression $p$ value is smaller than 0.05.

**GTEx eQTL analysis**. We downloaded eQTL loci across multiple tissues from the GTEx database (accession phs000424.v7.p2), remapped the coordinates from GRCh37/hg19 to GRCh38/hg38, and removed eQTL-TSS pairs with $p$ values larger than $1e-5$. To investigate whether eQTL-TSS pairs in B-type TADs have longer distances than those in A-type TADs, we selected those eQTL-TSS pairs with TSS located in B-type and A-type TADs, and then compared their log-transformed eQTL-TSS distances using Student's $t$ test. To further examine whether the eQTL-TSS distances become shorter as chromatin becomes more active, we focused on TADs that have distinct activities in different lineages. Take brain cortex and lymphoblastoid for instance, we compared their H3K27ac level using Cistrome and defined brain-specific active TADs as those with significantly ($p$ value $< 1e-10$) higher H3K27ac level in brain cortex compared with lymphoblastoid. Lymphoblastoid-specific active TADs were defined in the same way. Further for eQTL-TSS pairs with TSS located in brain-specific or lymphoblastoid-specific active TADs, we compared the log-transformed distances measured in brain cortex and lymphoblastoid using GTEx data.

For more-comprehensive comparisons, we compared all possible pairs from whole blood, stomach, lymphoblastoid, and brain cortex. We defined those pairs with eQTL-TSS distances longer in tissue-specific active TADs as non-significant.

**Hi-C analysis**. The Hi-C data in cool format was downloaded from ftp://cooler.csail.mit.edu/coolers/hg19/. We mapped contact pairs to B-type and A-type TADs, respectively, and compared the decreasing rate of contact frequencies with distances. Specifically, for given genomic distances ($d$), we calculated the average contact frequencies ($F$):

$$F(d) = \frac{\sum_{\text{all interacting pairs}(d)} \text{counts}}{|\text{all possible interaction pairs}(d)|}$$

**GWAS-SNP enrichment analysis**. We mapped the disease-associated SNPs to TADs and calculated the number of SNPs in each TAD. Relative TAD-wise TF abundance was calculated as following steps:

1. For each of 216 TF ChIP-seq samples, the top 20,000 peaks were included.
2. For each of 3051 TADs, numbers of TF peaks located within were calculated. The relative TAD-wise TF abundance was calculated by dividing TF peak number with the mean TF peak number.

Using the relative TAD-wise TF abundance to approximate the TF occupancies overlapping with SNPs, we can write the equation as:

$$\text{TFscore} = \sum_{\text{TADs}} \#\text{SNPs} \cdot (\text{relative TAD} - \text{wise TF abundance})$$

Which can be further approximated as cosine similarities:

$$\frac{\sum_{\text{TADs}} \#\text{SNPs} \cdot (\text{relative TAD} - \text{wise TF abundance})}{|\#SNPs|_2 \cdot |\text{relative TAD} - \text{wise TF abundance}|_2}$$

We plotted the diseases/traits–TF heatmap with cosine similarities between relative TF occupancies and number of disease SNPs in TADs, and clustered the diseases/traits and TFs using hierarchical clustering.

**Reporting summary**. Further information on research design is available in the Nature Research Reporting Summary linked to this article.

## Data availability

The data that support the findings of this study are available as follows: Cistrome [http://cistrome.org/db/#/], CCLE [https://portals.broadinstitute.org/ccle], GTEx [https://www.gtexportal.org/home/], Hi-C[40] [ftp://cooler.csail.mit.edu/coolers/hg19/], GWAS[41] [https://www.ebi.ac.uk/gwas/home], Linkage disequilibrium blocks[69] [http://distild.jensenlab.org/download.html], CAGE[39] [http://enhancer.binf.ku.dk/presets/]. All other relevant data supporting the key findings of this study are available within the article and its Supplementary Information files or from the corresponding authors upon reasonable request. Key processed data for Figs. 1e, 2a, 2b, 2c, 3h, and Supplementary Fig. 2d are provided as Supplementary Data. Data set used for figures in the current study are available as Source Data. A reporting summary for this Article is available as a Supplementary Information file.

## Code availability

Open source software is listed below: MACS2 2.1.2: https://github.com/taoliu/MACS. LiftOver: http://hgdownload.soe.ucsc.edu/admin/exe/macOSX.x86_64/liftOver. Homer v4.11: http://homer.ucsd.edu/homer/index.html. For inference of TF regulatory decay distance: https://bitbucket.org/liulab/tf_regulatory_distance/src/master/. Scripts specific for each figure are available with request to corresponding authors.

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

## Acknowledgements

We thank Drs. Leonid Mirny, Bing Ren, Fulai Jin, and Mike Pazin for the helpful discussions during manuscript preparation. We also acknowledge the following funding sources for supporting our work: National Key Research and Development Program of China 2017YFC0908500, NIH grant U24 HG009446, and U24 CA237617 to X.S.L. and C.M.

## Author contributions

C.-H.C and C.M. conceived the project. C.-H.C performed the analyses. R.Z., J.F., C.W., X.D. helped data collection. C.T., Q.T. M.B., and J.S.L. provided suggestions. C.M. and X. S.L supervised the whole study and wrote the manuscript with C.-H.C. with the help of all other authors.

## Competing interests

The authors declare no conflict of interest.
