## [Peer Review File · Nature Communications]

Reviewers' comments:

Reviewer #1 (Remarks to the Author):

The authors have done a good job of revising the text for clarity, and responding to the points raised in my review. Most of the revisions, such as the re-casting of the GWAS discussion, make the story more clear. A few concerns remain, as follows.

Review point 2: The supplementary information about statistical details underlying the approach should be called out more clearly in the main text. Furthermore, the approach itself should be revised for clarity. For example, in line 552, what is the justification for the claim that the Pearson correlation between a gene and TF is the maximum likelihood estimator of the coefficient $\gamma_{i,j}$? In general, the exposition in this new section is hard to follow. In particular, the logical relationship between equations (3) and (4) is not clear in the text. I think perhaps line 557 should start with a phrase like "We claim that" and then line 561 should say, "To prove this claim, note that $\hat{\gamma}_{i,j}$ can also be interpreted as the estimated transcription ..., which can be approximated as" But then we have a forward reference in line 565 to $\hat{\Delta}_i$, and an incorrect reference in line 566 to supp fig 1a (should be 2a). This sentence seems to provide empirical evidence in support of the claim. The text should be rewritten to make clear what the primary claim is, and what the steps are to support that claim theoretically (with justification for each). Empirical support for the claim should then come after the theoretical analysis.

line 92-94: This sentence should mention that ChIP-seq data is used in the calculation.

line 165: As pointed out in the review response, it's certainly possible that Figure 1e contains more than two modes, but even if this is the case the two-class analysis here would still be useful. This point should be made in the manuscript, either here or in the Discussion.

line 194: It seems strange to say that the correspondence between hot/cold and A/B cannot be confirmed. One could certainly check whether the distinction holds up in the cell types for which Hi-C data is available.

Reviewer #2 (Remarks to the Author):

Chen et al, have now addressed some (but not all) my initial concerns on the revised manuscript entitled "Determinants of transcription factor regulatory range". There is a clear improvement in the text and now the manuscript reads better. Still, there are some sections that are difficult to follow and I would suggest the authors to simplify it as possible. Nevertheless, my main concern still holding is the disentangle of confounding factors in the analysis, including the Hi-C data and chromosome compartments as well as TADs. Also, I find the re-classification of TAD into "hot" and "cold" TADs unnecessary when it is likely that the authors are describing "hot" TADs as those that are mostly in A compartment for most cell types and "cold" TADs as those that are mostly in B compartment in most cells. Why do we need new definitions? Next, I also outline some unresolved concerns:

- The authors indicate that since Hi-C is not available in many cells types; they won't do a full interaction analysis of their findings. I am sure that for many of the cell types studied, one could find a "deep enough" Hi-C dataset to test the A/B compartment association of hot and cold TADs at least for some cell lines. In fact, the "limited" availability of Hi-C datasets did not stop the authors in defining TADs based solely on one dataset (not clear which ones from Methods as Dixon et al provided several) and using the results in all the different cell lines used in the work.
- Unfortunately, the authors did not provide evidence that "the results are not sensitive to peak-

calling method". They do not provide an analysis of size and distance between peaks.

- My original question of "why a z-score of 1.0 is taken as threshold to that a TAD is targeted by a TF?" Is still valid and has not been addressed. In fact, the authors state in the response that this is "an arbitrary threshold, not implying statistical significance". As such, one should assess the impact of this choice in the results/conclusions.

- Whether RNA expression correlates with enhancer-promoter proximity is a big debate in the field of 4DNucleomics and there are many recent papers with contradictory findings. However, the fact that "Hi-C chromatin interaction frequencies between distant loci are similar in hot and cold TADs this suggests that direct chromatin interactions between enhancers and promoters need not necessarily occur for an enhancer to regulate a distant gene" (as indicated in the response to reviewers), does not add light to the controversy (in my opinion). First, hot and cold TADs were new definitions in this article based solely on the fact of one chromatin mark. Second, the TAD coordinates (borders) were defined in a single cell type and may (or may not) be conserved for the other cell types used to generate the hot-and-cold classification. Third, there is no analysis of the size and gene distribution in those hot and cold TADs that could change the interpretation of the results. Fourth, the analysis of Hi-C is based on the interactions genomic decay, which is stated by the authors is not significantly different between hot and cold TADs. However, no statistic is provided to show no differences in the plot of genomic decay, which is stopped at distance of 1Mb. I would suggest that the authors use the % of interactions vs full extend of genomic a distance (similar to those in doi/10.1038/nature23001 see Extended data-figure 4a) for both types of TADs and demonstrate that the distributions are not statistically significantly different.

Summary of revisions:

Thank reviewers for insightful comments, which have helped us to substantially improve our study. We have thoroughly addressed the reviewers' concerns, adding substantial new analysis to support the assumptions and to reaffirm the validity of our findings. The major changes to the revised manuscript include:

- Analysis of the relationship between “hot” and “cold” TADs and A and B compartments in 21 Hi-C data sets. This analysis shows that “hot” TADs overlap with A compartments and “cold” TADs overlap with B compartments to a high degree in these data sets. Following the reviewers' suggestions, in the revised manuscript we refer to the TADs with high levels of H3K27ac ChIP-seq signal across cell types as “A-type”, and the TADs with the low levels of H3K27ac ChIP-seq signal as “B-type.”
- Clarification of the regulatory distance modeling approach. We have clarified the methods section describing the model used to infer the regulatory distances of different transcription factors.
- Rationale and alternative method for using the regulatory distance model. We have supplied additional empirical data supporting the use of our model. In addition, we have used an alternative method for inferring regulatory distances, obtaining similar results as before.
- Analysis using ChIP-seq peaks called by a different peak caller. We analyzed ChIP-seq data using the Homer peak caller obtaining results consistent with the previously reported ones.
- Analysis using TADs defined from a different data set. Using a different TAD definition based on an analysis from an alternative cell type we obtained results consistent with our previously reported findings, which were based on the TADs defined by Dixon et al, (Nature, 2012).
- New figures comparing Hi-C decay rates with regulatory distances. We present figures that compare the interaction frequencies observed in Hi-C in A-type and B-type TADs with the regulatory distances inferred from our analysis. One of the rationales for the development and use of Hi-C technologies is that they are informative about cis-regulatory interactions. We believe that incorporating and comparing our results with Hi-C interaction trends is helpful for contextualizing our results.
- Analysis using different cutoffs to call TF target TADs to support that long-range TFs is more associated with H3K27ac and gene expression levels within their target TADs.

Most studies of transcriptional regulation focus on individual TFs. In this study we investigate properties that are held in common by several TFs. Based on these commonalities, we provide practical solutions for TF target identification and for the identification of disease-related TFs from GWAS SNPs. We reveal scientific insights on distinct TF regulatory distances and on the effects of chromatin states on gene regulation. This paper contributes to a simpler and more predictive framework for interpreting complex gene regulation.

In the document that follows, reviewer comments are in regular font followed by our response in blue italics.

Reviewers' comments:

Reviewer #1 (Remarks to the Author):

The authors have done a good job of revising the text for clarity, and responding to the points raised in my review. Most of the revisions, such as the re-casting of the GWAS discussion, make the story more clear. A few concerns remain, as follows.

We appreciate the reviewer's careful reading of the manuscript and constructive criticism.

Review point 2: The supplementary information about statistical details underlying the approach should be called out more clearly in the main text. Furthermore, the approach itself should be revised for clarity. For example, in line 552, what is the justification for the claim that the Pearson correlation between a gene and TF is the maximum likelihood estimator of the coefficient $y_{i,j}$? In general, the exposition in this new section is hard to follow. In particular, the logical relationship between equations (3) and (4) is not clear in the text. I think perhaps line 557 should start with a phrase like "We claim that" and then line 561 should say, "To prove this claim, note that $\hat{y}_{i,j}$ can also be interpreted as the estimated transcription ..., which can be approximated as" But then we have a forward reference in line 565 to $\hat{\Delta}_i^*$, and an incorrect reference in line 566 to supp fig 1a (should be 2a). This sentence seems to provide empirical evidence in support of the claim. The text should be rewritten to make clear what the primary claim is, and what the steps are to support that claim theoretically (with justification for each). Empirical support for the claim should then come after the theoretical analysis.

We have called out the supplementary information with the statistical details more clearly, and have made substantial revisions to the description of the approach in the supplementary methods section.

Re: "the justification for the claim that the Pearson correlation between a gene and TF is the maximum likelihood estimator of the coefficient $y_{i,j}$ ":

We assume that there is an observable association in gene expression between TFs and the genes they regulate. Under this assumption, for an activating factor, the directly regulated genes of a TF should be more highly expressed in samples where the TF is highly expressed than in samples where the TF is not expressed or expressed at a low level. A simple way to approximate the relationship between the gene expression of TF i and another gene j in sample k is using the linear relationship:

$$E_{j,k} = \alpha + \gamma_{i,j}E_{i,k} + \epsilon_j \quad (2)$$

where $E_{i,k}$ and $E_{j,k}$ are the standardized expression levels of TF i and gene j in sample (cell line or tissue) k , $\gamma_{i,j}$ is the transcriptional regulatory effect of TF i on gene j , and $\epsilon_j \sim N(0, \sigma^2)$. The result that the Pearson correlation is the maximum likelihood estimator of $\gamma_{i,j}$ under the assumptions of this model can be deduced using standard analysis, which we show in the supplemental methods section of the revised manuscript. The purpose of this model is to quantify the relationship between the expression of a TF and the genes it regulates. As TF activities are often regulated in post-transcriptional ways, including sequestration of the factor in

the cytoplasm and post-translational modification, it is possible that the relationship between gene expression levels of a TF and its target genes is not observable. We show empirically that, for many TFs, the estimator $\rho_{i,j}^{\text{expr}}$ is indeed significantly associated with ChIP-seq predicted transcription regulation (Supplementary Fig. 2a). As suggested, we distinguish the statistical derivation aspects from the empirical ones.

Re: “the logical relationship between equations (3) and (4) is not clear in the text”:

The relationship between equations (3) and (4) is related to the biological assumption that TF binding in relatively close proximity to genes influences the expression of those genes; equation (4) is not a direct logical consequence of equation (3). In equation (4) we assume that, for some value Δ_i , the ChIP-seq peak derived RP model, $R_{i,j}(\Delta_i)$, can approximate $\hat{y}_{i,j}$, the gene expression derived estimate of the regulatory effect of TF i on gene j :

$$\hat{y}_{ij} = f(R_{i,j}(\Delta_i^*)) + \epsilon_{ij} \quad (4)$$

where $f(\cdot)$ could be a linear or non-linear function. With this relationship, gene expression and ChIP-seq data is used to quantify what “close proximity” means. In the original manuscript we presented analysis based on a linear model. In the revised manuscript we show that our results are not highly sensitive to model choice and similar findings can be produced using statistics that characterize non-linear relationships.

In particular, we carry out an analysis using distance correlation, a measure of association that is independent of the underlying variable distributions². Viewing $R_{i,j}(\Delta)$ with different Δ as different models, we performed goodness of fit tests for model selection using the distance correlation (dCor) between observed data and assumed model.

$$\Delta_i^* = \operatorname{argmax}_{\Delta} (\text{dCor}(\hat{y}_{ij}, R_{i,j}(\Delta)))$$

The above analysis yields a result that is consistent with previous findings based on linear analysis (Fig. R1-2). Moreover, we have carried out the analyses on many TFs with multiple ChIP-seq data sets, with multiple gene expression data sets, using different selections of peaks, and using different methods of calling peaks. Based on these results, we are confident that the conclusions of this work are insensitive to the selection of data or statistical framework. We have revised the text accordingly.

Fig. R1: Representative TFs with short-range (left) and long-range (right) regulatory decay distances using distance correlation as goodness-of-fit measure.

Fig. R2: Distribution of regulatory decay distances using distance correlation as goodness-of-fit measure.

line 92-94: This sentence should mention that ChIP-seq data is used in the calculation.

We have modified the corresponding sentence as follows:

“To infer the gene regulatory characteristics of different TFs, we used the previously described regulatory potential model (RP) to describe the relationships between TF ChIP-seq derived target genes and their gene expression levels (Fig 1a).”

line 165: As pointed out in the review response, it's certainly possible that Figure 1e contains more than two modes, but even if this is the case the two-class analysis here would still be useful. This point should be made in the manuscript, either here or in the Discussion.

We have addressed this concern by adding the following sentence:

“Although it is possible that there are more than two modes, the following two-class analysis would still be useful in contrasting the properties of TFs with different influence ranges.”

line 194: It seems strange to say that the correspondence between hot/cold and A/B cannot be confirmed. One could certainly check whether the distinction holds up in the cell types for which Hi-C data is available.

Following the reviewer’s suggestion, we processed data from “A Compendium of Chromatin Contact Maps Reveals Spatially Active Regions in the Human Genome”³, which annotated A/B compartments in 21 cells or tissues from HiC. We further calculated the percentage of compartment A of each TAD cluster (Fig. R3). As expected, the cold TADs (the left-most cluster) has lowest A/B ratio, while the hot TADs (the right-most cluster) has the highest percentage of compartment A. However, it’s arguable that our approach can better characterize chromatin states since we used more than 1,000 H3K27ac samples, which cover orders of magnitude more samples than Hi-C at a higher resolution. We have added this plot to Fig 2b. Following the reviewers’ suggestions, in the revision, we refer to hot and cold TADs as A-type and B-type TADs to avoid new definitions. The A-type and B-type TADs in the revision are still defined using H3K27ac ChIP-seq data and we make this clear in the revision.

Fig. R3: Compartment A/B ratio in TAD clusters.

Reviewer #2 (Remarks to the Author):

Chen et al, have now addressed some (but not all) my initial concerns on the revised manuscript entitled "Determinants of transcription factor regulatory range". There is a clear improvement in the text and now the manuscript reads better. Still, there are some sections that are difficult to follow and I would suggest the authors to simplify it as possible.

We appreciate the reviewer's efforts in carefully reading our manuscript. We have addressed the remaining concerns in this revision.

Nevertheless, my main concern still holding is the disentangle of confounding factors in the analysis, including the Hi-C data and chromosome compartments as well as TADs. Also, I find the re-classification of TAD into "hot" and "cold" TADs unnecessary when it is likely that the authors are describing "hot" TADs as those that are mostly in A compartment for most cell types and "cold" TADs as those that are mostly in B compartment in most cells. Why do we need new definitions?

In the revised manuscript we have addressed this concern about the new definitions of "hot" and "cold" TADs, replacing these terms with "A-type" and "B-type" TADs.

Next, I also outline some unresolved concerns:

- The authors indicate that since Hi-C is not available in many cells types; they won't do a full interaction analysis of their findings. I am sure that for many of the cell types studied, one could find a "deep enough" Hi-C dataset to test the A/B compartment association of hot and cold TADs at least for some cell lines. In fact, the "limited" availability of Hi-C datasets did not stop the authors in defining TADs based solely on one dataset (not clear which ones from Methods as Dixon et al provided several) and using the results in all the different cell lines used in the work.

To address the issue of whether hot and cold TADs correspond to A and B compartments we processed data from "A Compendium of Chromatin Contact Maps Reveals Spatially Active Regions in the Human Genome"³, which annotated A/B compartments in 21 cells or tissues from Hi-C. We calculated the proportion of A compartment overlap with each TAD cluster (Fig. R3) finding, as expected, the cold TADs (the left-most cluster) have the lowest A proportion, while the hot TADs (the right-most cluster) have the highest. In the revision, following the reviewers' suggestion we refer to hot and cold TADs as A-type and B-type TADs, without changing the definition of these TADs and making explicit that these TADs are defined using H3K27ac ChIP-seq data.

Regarding the TAD definition, we used the TAD boundaries defined from "Topological Domains in Mammalian Genomes Identified by Analysis of Chromatin Interactions"⁴. We did not use other TAD definitions since it has been shown that TAD boundaries tend to be similar in different cell types^{4,5,3}, and our analysis does not require precise definitions of the TAD boundaries. To demonstrate this point, we downloaded TAD boundaries defined from Hi-C in endometrial microvascular endothelial cells (EMEC) (ENCODE, GSE105710) and redid related analyses (Figs R4-R12). We found all the following observations to hold in the new TAD boundary definition: TF occupancy to be higher in A-type TADs (Fig. R5); gene density gene expression and GC content to be highest in A-type TADs and longer transcripts to occur in B-type TADs (Fig. R6); the relative occupancy of YY1 to be higher in A-type TADs and TEAD1 relative occupancy to be higher in B-type TADs (Fig. R7); long-range TF expression to be more

highly correlated with H3K27ac in long-range TF target TADs (Fig. R8); long-range TF expression to be more highly correlated with gene expression in long-range TF target TADs (Fig. R9); relative occupancies of long-range TFs to be associated with TADs harboring the TF gene locus (Fig. R10); long-range TFs tend to have longer regulatory distances in B-type TADs (Fig. R11), and eQTL-TSS distances tend to be longer in B-type TADs (Fig. R12).

• Unfortunately, the authors did not provide evidence that “the results are not sensitive to peak-calling method”. They do not provide an analysis of size and distance between peaks.

In the previous version of this manuscript we already addressed this issue by showing that our analysis results were robust to peak numbers (10,000 vs 20,000 peaks). Doubling the number of peaks has the effect of reducing the average distance of the closest TF binding site to a TSS by approximately a half, yet has a minimal effect on the regulatory distance. Even though the use of different peak callers doesn't make as big a difference as calling 10k vs 20k peaks, following the reviewer's suggestion, we re-processed the ChIP-seq data from fastq files and applied another peak caller (Homer) to analyze the 5 long-range and the 5 short-range TFs demonstrated in Fig. 1d. The resulting regulatory distances are consistent with previous analysis.

• My original question of “why a z-score of 1.0 is taken as threshold to that a TAD is targeted by a TF?” Is still valid and has not been addressed. In fact, the authors state in the response that this is “an arbitrary threshold, not implying statistical significance”. As such, one should assess the impact of this choice in the results/conclusions.

We repeated the analyses for Figs. 2e-h using z-score thresholds of 0.5, 1, and 1.5. These analyses all produce results that are consistent with the finding that, in comparison with short-range TFs, the expression of long-range TFs is more associated with H3K27ac and gene expression levels within their target TADs (Fig. R15).

between long-range and short-range TFs.

• Whether RNA expression correlates with enhancer-promoter proximity is a big debate in the field of 4DNucleomics and there are many recent papers with contradictory findings. However, the fact that “Hi-C chromatin interaction frequencies between distant loci are similar in hot and cold TADs this suggests that direct chromatin interactions between enhancers and promoters need not necessarily occur for an enhancer to regulate a distant gene” (as indicated in the response to reviewers), does not add light to the controversy (in my opinion).

We respectfully disagree with the reviewer. If progress is to be made in understanding the mechanisms of gene regulation, different perspectives will be needed, especially if recent papers have contradictory findings. At least, we should be allowed to present our observations and our viewpoint on this issue.

First, hot and cold TADs were new definitions in this article based solely on the fact of one chromatin mark.

H3K27ac ChIP-seq was used to define genomic context as this mark has been found to be associated with active enhancers and promoters and has been shown to be associated primarily with the A compartment⁶. Molecular dynamics simulations based on chromatin state are predictive of compartment formation⁷. In the revision, following the reviewers’ suggestion we refer to hot and cold TADs as A-type and B-type TADs, without changing the definition of these TADs and making explicit that these TADs are defined using thousands of H3K27ac ChIP-seq data. We show that there are several differences between A-type and B-type TADs besides the presence of this mark. The new analysis (Fig. R3) showed that A-type TADs correspond to the A compartment and B-type TADs correspond to the B compartment in most cell types for which Hi-C is available. Whereas A and B compartments are defined on the basis of a specific analysis of a single data type and Hi-C data is available in few cell types, H3K27ac ChIP-seq is available in hundreds of cell types. Our characterization of A-type and B-type TADs takes into consideration more cell types than those with available Hi-C data and this is a motivation for using H3K27ac ChIP-seq instead of Hi-C. Although we are changing nomenclature because the reviewers have suggested that it is unnecessary to introduce new definitions, the purpose of these TAD classifications is to define chromatin context not to predict A and B compartments, even if H3K27ac ChIP happens to be predictive of these compartments.

Second, the TAD coordinates (borders) were defined in a single cell type and may (or may not) be conserved for the other cell types used to generate the hot-and-cold classification.

TADs have been observed to be fairly well conserved between cell types^{4,5,3}. To show our results are not highly sensitive to TAD definitions we carried out an analysis using TADs that were defined in a different study and found consistent results (Fig. R4-R12).

Third, there is no analysis of the size and gene distribution in those hot and cold TADs that could change the interpretation of the results.

It is likely that gene density is associated with compartment type and is therefore likely to be important in defining chromatin architecture including phenomena such as A and B compartmentalization. A and B compartments are no less interesting because they differ in terms of various genetic properties. To address the question of whether factors such as the mean chromatin state influences the range of TF regulation, independently of effects such as gene density, we describe the effect of changes in the H3K27ac chromatin landscape in the

same TADs but in different cell types. In addition, this was consistently observed in many TADs and in many tissues (Fig. 4). We found that the chromatin context is indeed playing a role that cannot be explained by features such as size and gene distribution.

Fourth, the analysis of Hi-C is based on the interactions genomic decay, which is stated by the authors is not significantly different between hot and cold TADs. However, no statistic is provided to show no differences in the plot of genomic decay, which is stopped at distance of 1Mb. I would suggest that the authors use the % of interactions vs full extend of genomic a distance (similar to those in doi/10.1038/nature23001 see Extended data-figure 4a) for both types of TADs and demonstrate that the distributions are not statistically significantly different.

The main difference between the plot suggested by the reviewer (Extended Fig 4a) and the ones we presented is the range of the x-axis. Extended Fig 4a shows the range to 134Mb, which is far beyond the size of TADs (and compartments) and ranges of TF regulation. Further we observe that short range chromatin interactions (<10kb) are not well described by the same power law exponents that characterize Hi-C contact decay rates at longer distances. Although these short-range effects could be experimental artifacts of Hi-C, an orthogonal data type based on DamC reveals a similar trend⁸ (Fig. R16), suggesting that local chromatin characteristics may be truly distinct from long-range ones.

The question we address is whether the longer decay distances observed in B-type TADs can be explained from Hi-C contact decay rates. Decay rates in Hi-C data are typically quantified using power law models. Therefore, to compare Hi-C derived decay rates with the RP based analysis presented here, we modified the RP model using the power-law equation and determined the decay constants of long-range TFs to lie in a range between -0.35 and -0.47. The model determines decay rates are steeper in A-type TADs (-0.4 ~ -0.5) than in B-type TADs (-0.3 ~ -0.45) (Fig. R17), consistent with what we described in the manuscript. These decay rates are smaller (in terms of absolute magnitude) than those observed in Hi-C data, which are well beyond -0.5 (Fig. R18 for H1 TADs and Fig. R19 for EMEC TADs). Although it is interesting to note the differences in A-type and B-type TAD Hi-C decay rates in the 250kb to 1Mb range, this range is beyond the regulatory range of most TFs. Therefore, it is unclear whether this type of B-type TAD compaction is related to gene regulation by TF binding or enhancers.

Our paper, while not resolving the debates within the chromatin interaction community, does offer a different perspective on gene regulation. Our analysis describes typical ranges of regulatory interactions and the dependency of these ranges on the chromatin environment. If Hi-C is informative about gene regulation the relevant range for examining interactions would appear to be <100kb. Although compaction effects can be seen in B-type TADs at longer ranges it is unclear how these effects link to gene regulation at shorter ranges. The analysis presented in this paper shows a phenomenon of long-range gene regulation that, when reconciled with HiC data, could yield insights into gene regulation.

Fig. R16: Scaling of contact probabilities measured in DamC, 4C and Hi-C from all 130 TetO and 91 TetO-CTCF viewpoints. Power-law fitting was performed between 15kb and 1Mb. (adapted from “DamC reveals principles of chromatin folding in vivo without crosslinking and ligation”⁸)

Fig. R17: Using the power-law distance model, the inferred decay constant of long-range TFs in B-type TADs is significantly larger than those in A-type TADs (*t*-test *p*-value = 0.0013).

Fig. R18: Contact frequency versus genomic distances in B-type TADs and A-type TADs using 4 Hi-C data. The blue line marks the decay rate with -0.5. TADs were defined by H1 Hi-C.

Fig. R19: Contact frequency versus genomic distances in B-type TADs and A-type TADs using 4 Hi-C data. The blue line marks the decay rate with -0.5. TADs were defined by EMEC Hi-C.

References:

1. ERIC - EJ670704 - Four Assumptions of Multiple Regression That Researchers Should Always Test., Practical Assessment, Research & Evaluation, 2002. Available at: <https://eric.ed.gov/?id=EJ670704>. (Accessed: 26th November 2019)
2. Székely, G. J., Rizzo, M. L. & Bakirov, N. K. Measuring and testing dependence by correlation of distances. *Ann. Stat.* **35**, 2769–2794 (2007).
3. Schmitt, A. D. *et al.* A Compendium of Chromatin Contact Maps Reveals Spatially Active Regions in the Human Genome. *Cell Rep.* **17**, 2042–2059 (2016).
4. Dixon, J. R. *et al.* Topological domains in mammalian genomes identified by analysis of chromatin interactions. *Nature* **485**, 376–380 (2012).
5. Dixon, J. R. *et al.* Chromatin architecture reorganization during stem cell differentiation. *Nature* **518**, 331–336 (2015).

6. Rao, S. S. P. *et al.* A 3D Map of the Human Genome at Kilobase Resolution Reveals Principles of Chromatin Looping. *Cell* **159**, 1665–1680 (2014).
7. Shi, G. & Thirumalai, D. Conformational heterogeneity in human interphase chromosome organization reconciles the FISH and Hi-C paradox. *Nat. Commun.* **10**, (2019).
8. Redolfi, J. *et al.* DamC reveals principles of chromatin folding in vivo without crosslinking and ligation. *Nat. Struct. Mol. Biol.* **26**, 471–480 (2019).

REVIEWERS' COMMENTS:

Reviewer #1 (Remarks to the Author):

The authors have done a good job of responding to my concerns. I have no further comments or suggestions at this time.

Reviewer #2 (Remarks to the Author):

Chen et al, have now addressed significantly all concerns I still had.

Summary of revisions:

We thank the reviewers for their insightful comments. The responses to the reviewers' comments are listed below:

Reviewer #1 (Remarks to the Author):

The authors have done a good job of responding to my concerns. I have no further comments or suggestions at this time.

We thank the reviewers for their constructive suggestions in the review process.

Reviewer #2 (Remarks to the Author):

Chen et al, have now addressed significantly all concerns I still had.

We thank the reviewers for their constructive suggestions in the review process.